# Statistical Query Hardness of Multiclass Linear Classification with Random Classification Noise

Ilias Diakonikolas [* 1]  Mingchen Ma [* 1]  Lisheng Ren [* 1]  Christos Tzamos [* 2]

## Abstract

We study the task of Multiclass Linear Classification (MLC) in the distribution-free PAC model with Random Classification Noise (RCN). Specifically, the learner is given a set of labeled examples $(x, y)$, where $x$ is drawn from an unknown distribution on $\mathbb{R}^d$ and the labels are generated by a multiclass linear classifier corrupted with RCN. That is, the label $y$ is flipped from $i$ to $j$ with probability $H_{ij}$ according to a known noise matrix $H$ with non-negative separation $\sigma := \min_{i \neq j} H_{ii} - H_{ij}$. The goal is to compute a hypothesis with small 0-1 error. For the special case of two labels, prior work has given polynomial-time algorithms achieving the optimal error. Surprisingly, little is known about the complexity of this task even for three labels. As our main contribution, we show that the complexity of MLC with RCN becomes drastically different in the presence of three or more labels. Specifically, we prove *super-polynomial* Statistical Query (SQ) lower bounds for this problem. In more detail, even for three labels and constant separation, we give a super-polynomial lower bound on the complexity of any SQ algorithm achieving optimal error. For a larger number of labels and smaller separation, we show a super-polynomial SQ lower bound even for the weaker goal of achieving *any* constant factor approximation to the optimal loss or even beating the trivial hypothesis.

## 1. Introduction

A multiclass linear classifier is any function $f : \mathbb{R}^d \to [k]$ of the form $f(x) = \operatorname{argmax}_{i \in [k]}(w_i \cdot x)$, where $w_i \in \mathbb{R}^d$ for all $i \in [k]$. (If the maximum is achieved by more than one indices, the tie is broken by taking the smallest index.). Multiclass Linear Classification (MLC)—the task of learning an unknown linear classifier from random labeled examples—is a textbook machine learning problem (Shalev-Shwartz & Ben-David, 2014), which has been extensively studied both theoretically and empirically (Platt et al., 1999; Hsu & Lin, 2002; Aly, 2005; Duan & Keerthi, 2005; Tewari & Bartlett, 2007; Kakade et al., 2008; Huang et al., 2011; Beygelzimer et al., 2019). MLC naturally arises in a range of critical applications, including face recognition (Lihong et al., 2009), cancer diagnosis (Panca & Rustam, 2017), ecological indicators (Bourel & Segura, 2018) and more. In all these settings, the number of labels is much larger than two—hence they cannot be modeled by binary linear classification. Moreover, MLC has important connections with modern deep learning architectures; the last layer of a neural network is typically a softmax function layer—a natural extension of MLC.

The sample complexity of MLC is fairly well-understood in the PAC model, even in the presence of noise. Specifically, standard arguments, see, e.g., (Shalev-Shwartz & Ben-David, 2014), give that $\operatorname{poly}(d, k, 1/\epsilon)$ samples information-theoretically suffice to achieve 0-1 error $\operatorname{opt} + \epsilon$, where $\operatorname{opt}$ is the optimal error achievable by any function in the class. Yet the computational complexity of this task has remained perplexing. In the realizable setting (i.e., in the presence of clean labels), the sample complexity of MLC is $\tilde{O}(dk/\epsilon)$ and the problem is solvable in polynomial-time via a reduction to linear programming (LP). Notably, the corresponding LP can be solved efficiently in the Statistical Query (SQ) model (Kearns, 1998), using a rescaled Perceptron algorithm (Dunagan & Vempala, 2004). Alas, the realizable setting is highly idealized—in most practical applications, some form of partial label contamination is unavoidable. It is thus natural to ask what is algorithmically possible in the presence of label noise.

If the label noise is adversarial (Haussler, 1992; Kearns et al., 1994) or even semi-random (Massart & Nédélec, 2006), strong computational hardness results are known even for *binary* linear classification, corresponding to $k = 2$ (Daniely, 2016; Diakonikolas et al., 2022; Diakonikolas & Kane, 2022; Nasser & Tiegel, 2022; Diakonikolas et al.,

---

*Equal contribution [1]Department of Computer Sciences, University of Wisconsin-Madison, Madison, USA [2]University of Athens and Archimedes AI, Athens, Greece. Correspondence to: Mingchen Ma <mingchen@cs.wisc.edu>, Lisheng Ren <lren29@wisc.edu>.

*Proceedings of the $42^{nd}$ International Conference on Machine Learning*, Vancouver, Canada. PMLR 267, 2025. Copyright 2025 by the author(s).

2023a; Tiegel, 2023). These hardness results, of course, carry over to the multiclass setting.

Interestingly, if the label noise is random—formalized by the Random Classification Noise (RCN) model (Angluin & Laird, 1988)—the binary linear classification task admits a polynomial-time algorithm. The first such algorithm was given in (Blum et al., 1998); see also (Dunagan & Vempala, 2004; Diakonikolas et al., 2021; 2023b; Kontonis et al., 2024). All these algorithms are known to fit the SQ model.

The preceding discussion motivates the algorithmic study of MLC in the presence of RCN. A positive algorithmic result for the multiclass case with RCN would be of significant theoretical and practical interest. *As our main contribution, we give strong evidence that such an efficient algorithm does not exist.*

To formally state our contributions, we require the definition of multiclass classification with RCN that has been widely studied in prior works (Patrini et al., 2017; Van Rooyen & Williamson, 2018; Ghosh et al., 2017).

**Definition 1.1** (Multiclass Classification with RCN). Let $X$ be the space of examples and let $Y = [k]$ be the label space. A multiclass classifier is any function $f : X \to Y$. A noise matrix $H \in [0,1]^{k \times k}$ is a row stochastic matrix such that for every $i \in [k]$, $\sum_{j=1}^{k} H_{ij} = 1$. An instance of multiclass classification with RCN is parameterized by $(D, f^*, H)$, where $f^*$ is the ground truth multiclass classifier, $H$ is the noise matrix, and $D$ is a joint distribution over $X \times Y$ such that each labeled example $(x, y) \sim D$ is generated as follows. We have $x \sim D_X$, where $D_X$ is the marginal of $D$ over $X$. The label $y$ of $x$ is drawn from the distribution such that $\mathbf{Pr}(y = j \mid x) = H_{f^*(x)j}$ for $j \in [k]$. The error of a hypothesis $h : X \to Y$ is defined as $\mathrm{err}(h) := \mathbf{Pr}_{(x,y) \sim D}(h(x) \neq y)$. Given a set $S$ of i.i.d. examples drawn from $D$, $\epsilon \in (0, 1)$, and a function class $\mathcal{F}$ such that $f^* \in \mathcal{F}$, a learner is asked to output a hypothesis $\hat{h}$ such that $\mathrm{err}(\hat{h}) \leq \mathrm{opt} + \epsilon$, where $\mathrm{opt} := \min_{f \in \mathcal{F}} \mathrm{err}(f)$.

We will also consider algorithms with approximate error guarantees, namely aiming for 0-1 error $C\mathrm{opt} + \epsilon$, where $C$ is a universal constant. While the focus of this paper is on the setting that $\mathcal{F}$ is the class of multiclass *linear* classifiers, the broader class of multiclass polynomial classifiers will arise in our proof. A multiclass degree-$m$ polynomial classifier $f_P : \mathbb{R}^d \to [k]$ is characterized by a collection $P = (p_1, \ldots, p_k)$, where $p_i(x) : \mathbb{R}^d \to \mathbb{R}, i \in [k]$, is a polynomial of degree at most $m$. For $x \in \mathbb{R}^d$, $f_P(x) = \mathrm{argmax}_{j \in [k]} p_j(x)$. (If the maximum is achieved by more than one indices, the tie is broken by taking the smallest index.)

If $\sigma := \min_{i \neq j} H_{ii} - H_{ij} \geq 0$, MLC with RCN can be solved up to error $\mathrm{opt} + \epsilon$ with sample complexity $\min\{\tilde{O}(dk/\sigma\epsilon), \tilde{O}(dk/\epsilon^2)\}$ via empirical risk minimiza-

tion (ERM). (Moreover, the ground truth $f^*$ achieves the optimal 0-1 loss of opt.) This can be directly deduced from (Massart & Nédélec, 2006). Though the sample complexity of the problem is understood, its computational complexity has remained open even for the case of 3 labels.

A long line of work (Wang et al., 2017; Patrini et al., 2017; Van Rooyen & Williamson, 2018; Lipton et al., 2018; Zhang et al., 2021) has focused on understanding MLC, and more general multiclass classification, with RCN from an algorithmic perspective—both theoretically and empirically. The methods proposed and analyzed in these works require inverting the noise matrix $H$ during the training process, and achieve sample and time complexity scaling inverse polynomially with the minimum singular value of $H$. This quantity could be arbitrarily small or even zero— even if $k = 3$ and separation $\sigma = 0.1$. Hence, such approaches do not lead to an efficient algorithm in general.

For binary classification, (Kearns, 1998) showed that any efficient SQ algorithm that succeeds in the realizable setting can be efficiently converted into an efficient SQ algorithm that solves the same problem in the presence of RCN. Unfortunately, no such result is known for the multiclass setting. This suggests that an efficient algorithm for MLC with RCN would require novel techniques.

Perhaps surprisingly, here we provide strong evidence that such an efficient algorithm does not exist, even for the case of 3 labels. Formally, we establish the first super-polynomial SQ lower bounds for MLC with RCN, suggesting that the complexity of the problem dramatically changes for $k \geq 3$.

SQ algorithms are a class of algorithms that, instead of having direct access to samples, are allowed to query expectations of bounded functions of the distribution (see Definition 3.1). The SQ model was introduced in (Kearns, 1998). Subsequently, the model has been extensively studied in a variety of contexts (Feldman, 2016). The class of SQ algorithms is broad and is known to capture a range of algorithmic techniques (Feldman et al., 2017a;b).

All our hardness results hold even if the noise matrix $H$ is known and given as input to the learner. Our first main result pertains to the case of optimal error guarantee, and holds even if $\sigma$ is a positive constant. In particular, we show:

**Theorem 1.2** (Informal Statement of Theorem 6.1). *There is a noise matrix $H \in [0,1]^{3 \times 3}$ with $H_{ii} - H_{ij} \geq 0.1, \forall i \neq j \in [3]$, such that it is SQ-hard to learn an MLC problem on $\mathbb{R}^d$, with RCN specified by $H$, up to error $\mathrm{opt} + \epsilon$.*

Given Theorem 1.2, it is natural to ask whether it is possible to *approximately* efficiently learning MLC under RCN. A learner is said to be $C$-approximate if it outputs a hypothesis $\hat{h}$ such that $\mathrm{err}(\hat{h}) \leq C\mathrm{opt} + \epsilon, \forall \epsilon \in (0, 1)$, where $C > 1$. By considering larger values of $k$ and small separation $\sigma$, we show that even approximate learning is SQ-hard (even if

the noise level is small). In more detail, we show:

**Theorem 1.3** (Informal Statement of Corollary 6.3). *For any $C > 1$, there exists a noise matrix $H \in [0,1]^{k \times k}$, where $k = O(C)$ and $\sigma = \min_{i,j} H_{i,i} - H_{i,j} = \Omega(1/C)$, such that it is SQ-hard to learn an MLC problem on $\mathbb{R}^d$ with RCN specified by $H$ up to error $C$opt, even if $\mathrm{opt} = \Theta(1/C)$.*

In fact, our final result shows that within the regime $\sigma = \min_{i \neq j} H_{ii} - H_{ij} > 0$, it is even SQ-hard to learn a hypothesis with error $1 - 1/k - o(1)$, even if $\mathrm{opt} = O(1/k)$. That is, it is hard to even find a hypothesis better than guessing the labels uniformly at random. Specifically, even if we only add 1% RCN for an instance of MLC, it is hard to learn a hypothesis with error better than 99% for $k = 100$.

**Theorem 1.4** (Informal Statement of Corollary 6.4). *For any $k \in \mathbb{Z}_+$ with $k \geq 3$, there is a noise matrix $H \in [0,1]^{k \times k}$ with $\sigma = \min_{i,j} H_{i,i} - H_{i,j} = \Omega(1/d)$ such that it is SQ-hard to learn an MLC problem on $\mathbb{R}^d$ with RCN specified by $H$ up to error $1 - 1/k - o(1)$, even if $\mathrm{opt} = O(1/k)$.*

## 2. Technical Overview

At a high level, our proof leverages the SQ lower bound framework developed in (Diakonikolas et al., 2017) and techniques for constructing distributions that match Gaussian moments from (Diakonikolas & Kane, 2022) and (Nasser & Tiegel, 2022). We stress that while these ingredients are useful in our construction, employing them in our context requires novel conceptual ideas.

Roughly speaking, prior work (Diakonikolas et al., 2017) and its generalization (Diakonikolas & Kane, 2022) give a generic framework for proving SQ-hardness results for supervised learning problems. Consider a distribution $D$ over $\mathbb{R}^d \times [k]$—a distribution consistent with an instance of a classification problem. Suppose that for $y \in [k]$, the distribution of $x$ conditioned on $y$ has the form of $P_v^A$, for $v$ a hidden direction in $\mathbb{R}^d$, such that: (i) $x_v$, the projection of $x$ on the $v$ direction, follows a one-dimensional distribution $A$, and (ii) $x_{v^\perp} \sim N(0, I)$. Moreover, suppose that the one-dimensional distribution $A$ nearly matches the first $t$ moments of $N(0,1)$, within error $\nu$, and has chi-squared norm at most $\beta$. The any SQ algorithm that correctly distinguishes between the case where $(x,y)$ is drawn from such a $D$ versus the case where the label $y$ is generated independently of $x$ according to $D_y$, needs either to make $2^{\Omega(d)}$ statistical queries or to make a query with tolerance $2\sqrt{\tau}$, where $\tau = \nu^2 + 2^{-\Omega(t)}\beta$.

To leverage the aforementioned result, it suffices for us to construct a distribution $D$ with the form discussed above that is consistent with an instance of MLC with RCN. Unfortunately, constructing such a $D$ directly is technically challenging. To overcome this obstacle, we first note that

one can reduce learning multiclass degree-$m$ *polynomial* classifiers with RCN to learning linear classifiers with RCN using the Veronese mapping, defined as $V(x) := (x,1)^{\otimes m}$. Therefore, it suffices for us to give a distribution $D$ that is consistent with an instance of a multiclass polynomial classifier $f^* = \mathrm{argmax}\{p_1(v \cdot x), \ldots, p_k(v \cdot x)\}$ with RCN over $\mathbb{R}^N$ instead. We remark that as we want to prove SQ hardness for MLC over $\mathbb{R}^d$ and $d = N^{O(m)}$, if $m$ is large, then we need to prove a stronger hardness result for the polynomial classification problem over $\mathbb{R}^N$. As we will discuss in Section 6, a key towards proving our SQ-hardness result is to choose the correct value of $m$.

Given the above discussion, it suffices to look at one-dimensional distributions along $v$. Our key observation, that leads to our SQ-hardness result, is that for a noise matrix $H$, if the $k$th row $h_k$ can be written as a convex combination $h_k = \sum_{j \in [k-1]} a_j h_j$ of the other rows, then an example drawn from the marginal distribution $\sum_{j \in [k-1]} a_j P_v^{A_j}$, where $P_v^{A_j}$ is the marginal distribution for $x$ with $f^*(x) = j$, will have observed label following the distribution $h_k$. Intuitively, if $D$ has such a marginal distribution of $x$, then the conditional distribution on $y \in [k]$ should be a mixture of the base distributions $P_v^{A_j}$. As long as $A_i, i \in [k-1]$ is close to $N(0,1)$, this in turn can be shown to imply a hardness result.

Such an intuition is useful but is not formally correct: in general, the conditional distribution is not exactly a mixture of the base distributions. We overcome this difficulty by mixing examples with $f^*(x) = i$ and $f^*(x) = k$ carefully to obtain $A_i, i \in [k-1]$. Specifically, we leverage techniques for constructing moment-matching distributions from (Diakonikolas & Kane, 2022; Nasser & Tiegel, 2022). Roughly speaking, these works considered the following distribution. Let $G_{\delta,\xi}$ be the distribution of $z \sim \mathcal{N}(0,1)$ conditioned on $z \in I_i = [i\delta - \xi, i\delta + \xi]$ for $i \in \mathbb{Z}_+$ i.e., conditioned on equally spaced intervals with width $\xi$. The distribution $G_{\delta,\xi}$ approximately matches moments up to degree $1/\delta^2$ up to error $2^{-\Omega(1/\delta^2)}$ with $N(0,1)$ and its chi-squared norm is not too large for any $\xi \geq 2^{-\Omega(1/\delta^2)}$. We choose $A_1 = G_{\delta,\xi}$. Inspired by (Nasser & Tiegel, 2022), for $i \in [-m, m]$, if we make tiny shifts for $A_1$ over these $I_i$, $k - 2$ times to get $k - 1$ unions of disjoint intervals $J_j, j \in [k-1]$, we obtain distributions $A_1, \ldots, A_{k-1}$ that are close to $N(0,1)$. Importantly, one can construct polynomials $p_j(z) > 0$ if and only if $z \in J_j, j \in [k-1]$, and $p_k(z) > 0$ if and only if $z \notin I_{in} := \bigcup_j J_j$. This implies that $D$ is consistent with an instance of multiclass polynomial classification with degree $O(m)$.

Moreover, we show that the larger $\mathbf{Pr}(z \in I_{in})$ is constructed, the better learning guarantee we can rule out. If we choose $\mathbf{Pr}(z \in I_{in}) = \epsilon$, then we are able to rule out

learning algorithms that achieve error $\text{opt} + \epsilon$. Moreover, by carefully designing the noise matrix $H$ and choosing $\mathbf{Pr}(z \in I_{in}) = 1 - 1/\text{poly}(k)$, we are additionally able to rule out algorithms with error better than $1 - 1/k$, even if $\text{opt} = O(1/k)$.

## 3. Preliminaries and Notations

Let $f^* : X \rightarrow Y$ be the ground truth hypothesis. For $j \in [k]$, denote by $S_j = \{x \mid f^*(x) = j\} \subseteq \mathbb{R}^d$ the set of examples with $f^*(x) = j$. Let $h : X \rightarrow Y$ be an arbitrary hypothesis. For $i, j \in [k]$, we denote by $S_{ji} = \{x \mid f^*(x) = j, h(x) = i\} \subseteq \mathbb{R}^d$, the set of examples with ground truth label $j$, but on which $h$ predicts $i$. In this paper, we use $\mathbb{S}^{d-1}$ to denote the unit sphere in $\mathbb{R}^d$. Let $K \subseteq \mathbb{R}^d$ be any set; we denote by **conv**$(K)$, the convex hull of $K$. For a noise matrix $H \in [0,1]^{k \times k}$, we denote by $h_i, i \in [k]$, the $i$th row vector of $H$.

For a distribution $D$, we use $\mathbf{E}_{\mathbf{x} \sim D}(x)$ to denote the expectation of $D$. Let $D$ be a distribution of $(x, y)$ over $\mathbb{R}^d \times [k]$. We use $D_X$ to denote the marginal distribution of $D$ over $\mathbb{R}^d$ and use $D_y$ to denote the marginal distribution of $D$ over $y$. In this paper, we will use $N(0, I)$ to denote the standard Gaussian distribution over $\mathbb{R}^d$ and use $N(0, 1)$ to denote the standard one-dimensional normal distribution. For $N(0, 1)$, we use $G(x)$ to denote its density function and use $\gamma_t, t \in \mathbb{N}$, to denote its standard $t$th moment $\mathbf{E}_{x \sim N(0,1)} x^t$.

**Definition 3.1** (SQ Model). Let $D$ be a distribution over $X \times Y$. A *statistical query* is a bounded function $q : X \times Y \rightarrow [-1, 1]$. We define $\text{STAT}(\tau)$ to be the oracle that given any such query $q$, outputs a value $v$ such that $|v - \mathbf{E}_{(x,y) \sim D}[q(x, y)]| \leq \tau$, where $\tau > 0$ is the *tolerance* parameter of the query. A *statistical query (SQ) algorithm* is an algorithm whose objective is to learn some information about an unknown distribution $D$ by making adaptive calls to the corresponding $\text{STAT}(\tau)$ oracle.

**Definition 3.2** (Pairwise Correlation). The pairwise correlation of two distributions with probability density function $D_1, D_2 : \mathbb{R}^d \mapsto \mathbb{R}_+$ with respect to a distribution with density $D : \mathbb{R}^d \mapsto \mathbb{R}_+$, where the support of $D$ contains the support of $D_1$ and $D_2$, is defined as $\chi_D(D_1, D_2) := \int_{\mathbb{R}^d} D_1(\mathbf{x}) D_2(\mathbf{x})/D(\mathbf{x}) d\mathbf{x} - 1$. Furthermore, the $\chi$-squared divergence of $D_1$ to $D$ is defined as $\chi^2(D_1, D) := \chi_D(D_1, D_1)$.

**Organization** The structure of this paper is as follows: In Section 4, we introduce an appropriate hypothesis testing problem and show how to efficiently reduce it to our learning problem. In Section 5, we construct the hard distribution for the testing problem. Finally, in Section 6, we prove the main results of this paper by carefully choosing the parameters and putting together the reduction of Section 4 and the hard distribution construction of Section 5.

## 4. From Hypothesis Testing to Learning

The usual way to prove an SQ-hardness result for a learning problem is to show hardness for an appropriate hypothesis testing problem that efficiently reduces to the learning problem. In this section, we explore the properties of multiclass classification problems and introduce the hypothesis testing problem. To start with, we give the following condition, which will be used to create a hard hypothesis testing problem related to the multiclass classification problem.

**Definition 4.1** (SQ-Hard to Distinguish Condition). Let $I = (D, f^*, H)$ be an instance of multiclass classification with RCN. We say that the instance $I$ satisfies the hard-to-distinguish condition if it satisfies the following conditions:

1. There exist $a_i \geq 0, i \in [k-1], \sum_{i=1}^{k-1} a_i = 1$ such that $h_k = \sum_{i=1}^{k-1} a_i h_i$.
2. $\forall i \in [k-1], \mathbf{Pr}_{x \sim D_X} (x \in S_i \mid x \neq S_k) = a_i$.
3. $H_{jj} - H_{ji} \geq 0, \forall j, i \in [k]$.

We give some intuition behind Definition 4.1. Given a noise matrix $H$, the first condition, Item 1, implies that the $k$th row vector of $H$ can be written as a convex combination of $h_i, i \in [k-1]$, with convex coefficients $a_i$. Recall that an example $x$ with ground truth label $i$ has $H_{ij}$ probability to be observed as label $j$. If the probability mass of $S_1, \ldots, S_{k-1}$ are assigned proportionally to these convex coefficients (Item 2), then a random example drawn from these regions will have probability $H_{ki}$ to have label $i$. Specifically, if we draw a random example $x \sim D_X$, then

$$
\begin{aligned}
&\mathop{\mathbf{Pr}}_{(x,y) \sim D} (y = i) \\
&= \mathop{\mathbf{Pr}}_{x \sim D_X} (x \notin S_k) \textstyle\sum_{j=1}^{k-1} a_j H_{ji} + \mathop{\mathbf{Pr}}_{x \sim D_X} (x \in S_k) H_{ki} \\
&= \mathop{\mathbf{Pr}}_{x \sim D_X} (x \notin S_k) H_{ki} + \mathop{\mathbf{Pr}}_{x \sim D_X} (x \in S_k) H_{ki} = H_{ki}.
\end{aligned}
$$

This implies that the marginal distribution $D_y$ follows the discrete distribution $h_k$. This suggests that given a set of examples drawn i.i.d. from $D$, without exploring the structure of the data, it is hard to tell whether the labels $y$ are generated from an instance of multiclass classification or generated from the distribution $h_k$ independent of $x$.

Motivated by this observation, we define the following testing problem, which we will show later to be hard to solve if some additional distributional assumptions are satisfied.

**Definition 4.2** (Correlation Testing Problem). A correlation testing problem $\mathcal{B}(D_0, \mathcal{D})$ is defined by a distribution $D_0$ and a family of distributions $\mathcal{D}$ over $X \times Y$. An algorithm is given SQ query access to some distribution $D$ and a noise matrix $H$, and is asked to distinguish whether $D = D_0$ or $D \in \mathcal{D}$. In particular, $D_0$ and $\mathcal{D}$ satisfy the following properties.

1. Null hypothesis: $x \sim (D_0)_X$ for some known distribution $(D_0)_X$, where $y$ is independent of $x$ such that

$\mathbf{Pr}_{y \sim (D_0)_y}(y = i) = H_{ki}, \forall i \in [k]$.

2. Alternative hypothesis: $D \in \mathcal{D}$, where each distribution $D \in \mathcal{D}$ corresponds to a multiclass classification instance $(D, f^*, H)$ that satisfies Definition 4.1.

The correlation testing problem asks an algorithm to test whether the distribution of $y$ is generated according to an instance of a multiclass classification problem or is generated from a known distribution that is independent of $x$. In the rest of this section, we establish the connection between the testing problem of Definition 4.2 and the multiclass classification problem. We first present the following error-decomposition lemma, that describes the error of any multiclass hypothesis $h$. We defer the proof of Lemma 4.3 to Appendix B.1.

**Lemma 4.3.** *Let $(D, f^*, H)$ be any instance of multiclass classification with RCN. Let $h : X \to Y$ be an arbitrary multiclass hypothesis over $X$. Then, $\mathrm{err}(h) = \sum_{j=1}^{k} \mathbf{Pr}(S_j)(1 - H_{jj}) + \sum_{i \neq j} \mathbf{Pr}(S_{ji})(H_{jj} - H_{ji})$ . In particular, if $H_{jj} - H_{ji} \geq 0$ for every $j \in [k], i \neq j$, then $\mathrm{opt} = \mathrm{err}(f^*) = \sum_{j=1}^{k} \mathbf{Pr}(S_j)(1 - H_{jj})$*

Given Definition 4.2, the following lemma reduces the correlation testing problem to the classification problem.

**Lemma 4.4.** *Let $\mathcal{D}$ be a family of distribution over $X \times Y$ such that each distribution $D \in \mathcal{D}$ corresponds to a multiclass classification instance $(D, f^*, H)$ that satisfies Definition 4.1. If there is an SQ learning algorithm $\mathcal{A}$ such that for every instance $(D, f^*, H), D \in \mathcal{D}$, $\mathcal{A}$ makes $q$ queries, each of tolerance $\tau$, and outputs a hypothesis $\hat{h}$ such that $\mathrm{err}(\hat{h}) \leq \mathrm{opt} + \alpha$, where $2\alpha = \sum_{j=1}^{k-1} \mathbf{Pr}(S_j)(H_{jj} - H_{jk})$, then there is an SQ algorithm $\mathcal{A}'$ that solves the correlation testing problem defined in Definition 4.2 by making $q + 1$ queries, each of tolerance $\min(\tau, \alpha/2)$.*

We defer the full proof of Lemma 4.4 to Appendix B.2 and give an overview of the proof below. Recall that the goal of the correlation testing problem is to tell whether the label $y$ is generated according to the discrete distribution $h_k$ independent of $x$ (null hypothesis) or is generated according to some $D$ from $\mathcal{D}$ (alternative hypothesis). In the former case, no hypothesis will have an error better than the constant hypothesis $h(x) \equiv k$, which has an error of $1 - H_{kk}$. In the latter case, by Lemma 4.3, we can show that $\mathrm{err}(k) - \mathrm{opt} = 2\alpha$. This implies that, if we can learn the multiclass classification problem up to error $\mathrm{opt} + \alpha$, then we only need to make a single query with tolerance $\alpha/2$ to check whether the hypothesis $\hat{h}$ we learn has error less than $1 - H_{kk} - \alpha/2$ to solve the testing problem.

Given Lemma 4.4, we briefly explain how we will use it to prove the hardness results. Notice that if we choose $H_{jj} - H_{jk} \geq c$, for some constant $c > 0$ and $j \in [k-1]$, then $\alpha = \Theta(1 - \mathbf{Pr}(S_k))$. Thus, Lemma 4.4 suggests that in this case,

to prove the hardness of the learning problem, it suffices for us to construct instances of a multiclass classification problem that satisfies Definition 4.1 and $1 - \mathbf{Pr}(S_k)$ is as large as the desired accuracy $\alpha$. In particular, the larger $1 - \mathbf{Pr}(S_k)$ can be constructed, the stronger hardness result we are able to obtain. We defer the details to Section 5.

## 5. Hardness of Hypothesis Testing

Given Lemma 4.4, we know that it suffices to construct a correlation testing problem $\mathcal{B}(D_0, \mathcal{D})$ that is hard to solve. Notice that any multiclass polynomial classification problem can be represented as a multiclass linear classification problem in a higher dimension via the kernel method. So, we will consider constructing a family of correlation testing problems $\mathcal{B}(D_0, \mathcal{D})$ for multiclass polynomial classification problems. Consider an instance $I = (D, f^*, H)$ of multiclass polynomial classification problem with degree-$m$ under the SQ-hard to distinguish condition, where $f^*$ is characterized by $k$ degree-$m$ polynomials of the form $p_1(v \cdot x), \ldots, p_k(v \cdot x), v \in \mathbb{S}^{d-1}$. Then the label $y$ is completely dependent on $v$, which implies that a learner must look at examples close to $v$ in order to solve the testing problem. This motivates us to look at the following hidden direction distribution that is frequently used in the literature of SQ lower bounds (Diakonikolas et al., 2017).

**Definition 5.1** (Hidden Direction Distribution). Let $A$ be a distribution over $\mathbb{R}$, with probability density $A(x)$ and $v \in \mathbb{R}^d$ be a unit vector. Define $P_v^A(x) = A(v \cdot x) \exp\left(-\|x - (v \cdot x)v\|_2^2 / 2\right) / (2\pi)^{(d-1)/2}$, i.e., $P_v^A$ is a product distribution whose orthogonal projection onto $v$ is $A$ and onto the subspace orthogonal to $v$ is the standard $(d-1)$-dimensional Gaussian distribution.

Based on the definition of the hidden direction distribution, we construct the family of hard distributions $\mathcal{D}$ as follows.

**Definition 5.2** (Hidden Direction Distribution Family). Let $H \in [0, 1]^{k \times k}$ be a noise matrix that satisfies (1), (3) in Definition 4.1 with the convex combination coefficients $a \in [0, 1]^{k-1}$. Let $A = (A_1, \ldots, A_{k-1})$ be a list of $k - 1$ *base distributions*, where for $i \in [k - 1]$, $A_i$ is a one-dimensional distribution that satisfies the following conditions:

1. $\exists$ a set of $m$ disjoint intervals $J_i, i \in [k-1]$, such that $A_i(x) > 0$, for $x \in J_i$ and $A_i(x) = 0$, for $x \in I_{in} \setminus J_i$, where $I_{in} = \bigcup_{j \in [k-1]} J_j$.
2. $\forall x \in \mathbb{R} \setminus I_{in}, A_i(x) = A_j(x), \forall i, j \in [k-1]$.

We define the hidden direction distribution family $\mathcal{D} = \{D_v^{A,a}\}_{v \in \mathbb{S}^{d-1}}$ over $\mathbb{R}^d \times [k]$ such that $(x, y) \sim D_v^{A,a}$ is sampled as follows. With probability $a_j$, $x \sim P_v^{A_j}$. If $x \in J_j$, sample $y = i$ with probability $H_{ji}$, otherwise, sample $y = i$ with probability $H_{ki}$.

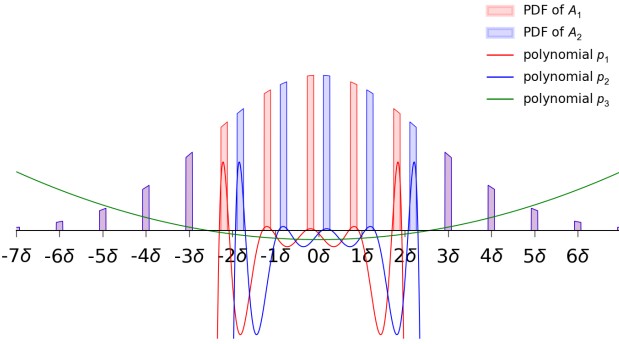

Figure 1: Illustration of base distributions for $k = 3$. Histograms that are colored in red (resp. blue) correspond to distribution $A_1$ (resp. $A_2$). $p_1, p_2, p_3$ colored in red, blue, and green are polynomials that characterize the target hypothesis $f^*$. $J_1$(resp. $J_2$) are red (resp. blue) intervals within the range $(-2\delta, 2\delta)$, where examples have ground truth label 1 (resp. 2). Examples outside $J_1 \cup J_2$ have ground truth label 3.

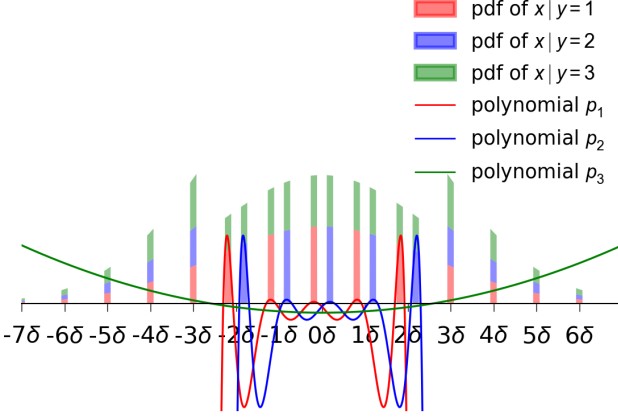

Figure 2: Illustration of $(D_v^{A,a})_{|y=i}$ for $k = 3$. $p_1, p_2, p_3$ colored in red, blue, green are polynomials that characterize the ground truth $f^*$. Histograms in red (resp. blue, green) correspond to distribution $(D_v^{A,a})_{|y=1}$ (resp. $(D_v^{A,a})_{|y=2}$, $(D_v^{A,a})_{|y=3}$). For each $i$, $(D_v^{A,a})_{|y=i}$ has many moments close to the moments of a standard normal.

We summarize the key properties of a hidden direction distribution family in Theorem 5.3, the main theorem of this section. Due to space limitations, the full proof is deferred to Appendix C.2.

**Theorem 5.3.** *Let $\mathcal{B}(D_0, \mathcal{D})$ be a correlation testing problem, where $(D_0)_X = N(0, I)$ and $\mathcal{D}$ is a hidden direction distribution family. Suppose there exists $\nu > 0$ such that for $\ell \leq t \in \mathbb{Z}_+$, the family of one-dimensional distribution $A_1, \ldots, A_{k-1}$ satisfies $\left|\mathbf{E}_{x \sim A_i} x^\ell - \gamma_\ell\right| \leq \nu$. Then:*

1. *Every distribution $D_v^{A,a} \in \mathcal{D}$ is consistent with an instance of multiclass polynomial classification with RCN $(D_v^{A,a}, f^*, H)$ with degree at most $2m$ that satisfies Definition 4.1.*
2. *For any small enough constant $c > 0$, let $\beta = \max_{i,j} \chi_{N(0,1)}(A_i, A_j)$ and let $\tau := \nu^2 + c^t\beta$. Any SQ algorithm that solves $\mathcal{B}(D_0, \mathcal{D})$ must make a query with accuracy better than $2\sqrt{\tau}$ or make $2^{\Omega_c(d)}\tau/\beta$ queries.*

In the rest of this section, we give an overview of the construction of the hidden direction distribution family as well as the proof of Theorem 5.3.

Consider a distribution $D_v^{A,a} \in \mathcal{D}$ (see Figure 1 for an example), where $\mathcal{D}$ is some hidden direction distribution family. Item 1 in the construction of a hidden direction distribution family is to ensure each $D_v^{A,a}$ is consistent with some multiclass polynomial classification problem with degree $O(m)$. Since for each $i \in [k-1]$, $J_i$ is a set of $m$ disjoint intervals, we know there is a degree-$2m$ polynomial $p_i(t) : \mathbb{R} \to \mathbb{R}$ such that $p_i(t) > 0$ if and only if $t \in J_i$. On the other hand, since $I_{in} = \mathbf{conv} \bigcup_{j \in [k-1]} J_j$ is a finite interval, there is a degree-2 polynomial $p_k(t) : \mathbb{R} \to \mathbb{R}$ such that $p_k(t) > 0$ if and only if $t \notin I_{in}$. Since

$J_i \cap J_j = \emptyset, \forall i \neq j$, we know that for each $j \in [k-1]$, if $v \cdot x \in J_j$, then $j = \text{argmax}\{p_1(v \cdot x), \ldots, p_k(v \cdot x)\}$ and if $v \cdot x \notin I_{in}, k = \text{argmax}\{p_1(v \cdot x), \ldots, p_k(v \cdot x)\}$. Thus, $D_v^{A,a}$ is consistent with an instance of multiclass polynomial classification with RCN $(D_v^{A,a}, f^*, H)$, where the marginal distribution is $\sum_{j=1}^{k-1} a_j P_v^{A_j}$ and the ground truth hypothesis $f^*(x) = \text{argmax}\{p_1(v \cdot x), \ldots, p_k(v \cdot x)\}$. This gives an overview of the first part of Theorem 5.3.

We next focus on the second part of Theorem 5.3. To simplify the notation, for each $i \in [k]$, we denote by $D_v^i = D_v^{A,a}(x \mid y = i)$ in the rest of this section. The proof strategy here is to use the standard SQ dimension (Lemma A.4). It is well-known that for any small constant $c$, there exists at least $2^{\Omega_c(d)}$ many unit vectors $u, v$ such that $|u, v| \leq c$ (see Fact A.5). Thus, to use Lemma A.4, we only need to bound $\chi_{D_0}(D_v^{A,a}, D_u^{A,a})$, where $|u \cdot v| \leq c$ as well as $\chi_{D_0}(D_v^{A,a}, D_v^{A,a})$. Since $\chi_{D_0}(D_v^{A,a}, D_u^{A,a}) = \sum_{i=1}^{k} H_{ki} \chi_{N(0,I)}\left(D_v^i, D_u^i\right)$, it is equivalent to upper bounding $\chi_{N(0,I)}\left(D_v^i, D_u^i\right)$. However, even though the base distributions $P_v^{A_j}$ are all close to $N(0, I)$ (thus have a small pairwise correlation), in general, conditional on the label $y = i$, there is no such guarantee for $D_v^i$. Here, we make use of Item 2, in the construction of the distribution family. Under Item 2, every $D_v^i$ is indeed a mixture of the distributions $P_v^{A_j}, j \in [k-1]$. Formally, we have the following technical lemma (see Figure 2 for intuition), the proof of which is deferred to Appendix C.1.

**Lemma 5.4** (Distribution Projection). *Let $\mathcal{D}$ be a hidden direction distribution family over $\mathbb{R}^d$ and let $D_v^{A,a} \in \mathcal{D}$ be a distribution that is consistent with an instance of multiclass polynomial classification with RCN $(D_v^{A,a}, f^*, H)$. For*

*every* $i \in [k]$, $D_v^{A,a}(x \mid y = i) = \sum_{j=1}^{k-1} \frac{a_j H_{ji}}{H_{ki}} P_v^{A_j}(x)$.

Given Lemma 5.4, to upper bound the pairwise correlation, it is equivalent to upper bound $\chi_{N(0,I)}(P_v^{A_j}, P_u^{A_j})$. Using the correlation lemma developed in (Diakonikolas & Kane, 2022), as long as each $A_j$ has many moments close to those of a standard normal distribution, $\chi_{N(0,I)}(P_v^{A_j}, P_u^{A_j})$ is small and we are able to prove hardness result using Lemma A.4. This gives an overview of the second part of Theorem 5.3.

**Construction of Hard Distributions**  In the rest of the section, we will construct the family of hard distributions $A_1, \ldots, A_{k-1}$. By Theorem 5.3, we require $A_1, \ldots, A_{k-1}$ to be supported in disjoint intervals and have many moments close to those of a standard normal distribution. The most natural construction of one such distribution is to restrict $N(0,1)$ over a sequence of discrete intervals. Related ideas have been used in proving hardness result of various learning problems, such as (Bubeck et al., 2019; Diakonikolas & Kane, 2022; Nasser & Tiegel, 2022; Tiegel, 2024). By choosing such a distribution as $A_1$, we are able to construct $A_2, \ldots, A_{k-1}$ by shifting the intervals constructed in $A_1$ with different step sizes. Such a construction can be viewed as a generalization of the technique in (Nasser & Tiegel, 2022) for proving SQ-hardness of learning a halfspace under Massart noise. We present the formal construction of our hard distributions in Definition 5.5 and list its properties in Proposition 5.6. We defer the proof of Proposition 5.6 to Appendix C.3.

**Definition 5.5.** For $\delta, \xi > 0$ such that $\delta > 4(k-1)\xi, 4(k-1)\xi < 1$, we define

$$G_{\delta,\xi} = \sum_{n \in \mathbb{Z}} \frac{\delta}{2\xi} G(x) \mathbb{1}(x \in [n\delta - \xi, n\delta + \xi])$$

and $G_{\delta,\xi}^{(n)} = G_{\delta,\xi} / \|G_{\delta,\xi}\|_1$, where $G(x)$ is the density of $N(0,1)$. We define $A_1(x) = G_{\delta,\xi}^{(n)}$ and for $i \in [k-1]$, we define

$$A_i(x) = \begin{cases} A_1(x + 4(i-1)\xi) & |x| \le m\delta + (4i-3)\xi \\ A_1(x) & |x| > m\delta + (4i-3)\xi, \end{cases}$$

where $m \in \mathbb{Z}_+$.

**Proposition 5.6.** *The univariate distributions* $A_1, \ldots, A_{k-1}$ *constructed in Definition 5.5 satisfy*

1. *$\exists$ a set of $m$ disjoint intervals $J_i, i \in [k-1]$ such that $A_i(x) > 0$, for $x \in J_i$ and $A_i(x) = 0$, for $x \in I_{in} \setminus J_i$, $I_{in} = \textbf{conv} \bigcup_{j \in [k-1]} J_j$.*
2. *$\forall x \in \mathbb{R} \setminus I_{in}, A_i(x) = A_j(x), \forall i, j \in [k-1]$.*
3. *For $i, j \in [k-1]$, $\chi_{N(0,1)}(A_i, A_j) \le O(\delta/\xi)^2$.*
4. *For $t \in \mathbb{N}$ and for $i \in [k-1]$, $|\mathbf{E}_{x \sim A_i} x^t - \gamma_t| \le O(t!) \exp(-\Omega(1/\delta^2)) + 4(k-1)\xi(1+2m\delta)^t$.*

## 6. Hardness of Multiclass Linear Classification Under RCN

In this section, we present our main hardness results for MLC. The proofs in this section use Lemma 4.4 to reduce the correlation testing problem (Definition 4.2) to the MLC learning problem, and construct a hidden direction distribution family (Definition 5.2) via the hard distribution defined in Definition 5.5 for the correlation testing problem. We will carefully choose the parameters for the hard distribution $A_1, \ldots, A_{k-1}$ to invoke Proposition 5.6 and Theorem 5.3 to get SQ lower bounds for different learning guarantees.

Here we first present the intuition behind the proof of our hardness results. As we discussed in Section 5, to prove the hardness result, it is sufficient to construct a hidden direction distribution family $\mathcal{D} = \{D_v^{A,a}\}_{v \in \mathbb{S}^{N-1}}$ consistent with a multiclass polynomial classification instance in $\mathbb{R}^N$ such that the base distributions $A_1, \ldots, A_{k-1}$ have many moments that are close to those of a standard normal. Recall that the construction of a hidden direction distribution family relies on a noise matrix $H$ that satisfies the SQ-hard to distinguish condition (Definition 4.1). As an example, consider the noise matrix

$$H = \begin{pmatrix} 0.6 & 0 & 0.4 \\ 0 & 0.6 & 0.4 \\ 0.3 & 0.3 & 0.4 \end{pmatrix}, \tag{1}$$

with $k = 3$, where $h_3 = (h_1 + h_2)/2$. Therefore, we choose base distributions $A_1, A_2$ constructed in Definition 5.5, $a = (1/2, 1/2)$ and $f^*(x) = \text{argmax}\{p_1(x), p_2(x), p_3(x)\}$ illustrated in Figure 1. Recall that the goal of the correlation testing problem is to tell whether the label $y$ is generated according to the discrete distribution $h_k$ or is generated by some $D_v^{A,a}$. By Lemma 4.4, for every $D \in \mathcal{D}$ $2\alpha := \text{err}_D(k) - \text{opt} = \sum_{j=1}^{k-1}(H_{jj} - H_{jk}) \mathbf{Pr}(S_j)$, the probability mass of $I_{in}$, the intervals in the middle as shown in Figure 2, with respect to $A_1$. This implies that the larger $\alpha$ is chosen, the better learning guarantee we can rule out. In particular, since $H_{jj} - H_{jk}, j \in [k-1]$ is larger than some universal constant, $\alpha$ is proportional to $\sum_{j=1}^{k-1} \mathbf{Pr}(S_j)$. By the construction of $A_1, \ldots, A_{k-1}$, this quantity is exactly $\mathbf{Pr}_{z \sim A_1}(z \in I_{in}) = \mathbf{Pr}_{z \sim A_1}(z \in I_{in})(|z| \le (m+1)\delta)$. Since $A_1$ is an approximation of a standard normal, the parameters $m, \delta$ of $A_1$ are selected such that $\mathbf{Pr}_{z \sim N(0,1)}(|z| \le m\delta) \propto \alpha$. On the other hand, by Proposition 5.6, for a given pair of $m, \delta$, by properly choosing small $\xi$, one can make the accuracy parameter $\tau$ in Theorem 5.3 as small as $\exp(-\text{poly}(1/\delta))$. However, this does not imply that we can choose $\delta$ arbitrarily small for the following reason: to solve the polynomial classification problem in $\mathbb{R}^N$, we need to embed the instance to $\mathbb{R}^d, d = N^{O(m)}$, and solve it with an algorithm for MLC. Therefore, if $\delta$ is chosen too small, $m$ could be too large to rule out any hardness result for MLC. That is, a good

tradeoff between $m, \delta$ is needed to prove our hardness result.

**Hardness of Getting Error** $\mathrm{opt} + \epsilon$   Recall that for binary linear classifiers, there is an algorithm that runs in time $\mathrm{poly}(d, 1/\epsilon)$ and outputs a hypothesis with error $\mathrm{opt} + \epsilon$. Moreover, the algorithm works even when $\min_{i \neq j} H_{ii} - H_{ij} = 0$. As it turns out, this is not the case for the multiclass case. Our first SQ-hardness result shows that even if $k = 3$ and $\min_{i,j} H_{ii} - H_{ij} > c$ for some constant $c$, to learn a hypothesis up to error $\mathrm{opt} + \epsilon$, one needs super-polynomial SQ complexity. Formally, we establish the following Theorem 6.1, whose proof is deferred to Appendix D.1.

**Theorem 6.1.** *There is a matrix $H \in [0,1]^{3 \times 3}$ with $H_{ii} - H_{ij} \geq 0.1, \forall i \neq j \in [3]$, such that any algorithm $\mathcal{A}$ that distribution-free learns multiclass linear classifiers with RCN specified by $H$ on $\mathbb{R}^d$ to error $\mathrm{opt} + \epsilon$, $\epsilon \in (0,1)$, requires either (a) at least $d^{\tilde{\Omega}(\log^{0.98}(d)/\epsilon^{1.98})}$ queries, or (b) a query of tolerance at most $1/d^{\tilde{\Omega}(\log^{0.98}(d)/\epsilon^{1.98})}$.*

The proof of Theorem 6.1 follows the above intuition. To show that learning up to error $\mathrm{opt} + \epsilon$ is hard, we choose parameters $m, \delta$ such that $\mathbf{Pr}(I_{in}) \approx \epsilon$. By the concentration properties of $N(0,1)$, we only need to choose $m\delta \approx \epsilon$. In the proof, we show that $m = \epsilon\sqrt{N}$ suffices to give a super-polynomial lower bound.

**Hardness of Approximation and Beating Random Guess** Given the hardness result in Theorem 6.1 of getting error $\mathrm{opt} + \epsilon$, one natural question is what kind of error guarantee we can efficiently achieve for MLC. For larger values of $k$ and small separation $\min_{i \neq j} H_{ii} - H_{ij}$, we show it is also hard to get any constant factor approximation, or even find a hypothesis with an error nontrivially better than a random guess given $\mathrm{opt} = O(1/k)$. Formally, we first give the following theorem, whose proof is deferred to Appendix D.2.

**Theorem 6.2.** *For any $k \in \mathbb{Z}_+$ and $k \geq 3$, there is a noise matrix $H \in [0,1]^{k \times k}$ such that $\max_{i,j} H_{i,i} - H_{i,j} = \zeta > 0$ and has the following property: For any sufficiently large $d \in \mathbb{Z}_+$, any SQ algorithm $\mathcal{A}$ that distribution-free learns multiclass linear classifiers with RCN specified by $H$ on $\mathbb{R}^d$ to error $1 - 1/k - \zeta - 2\mu$ requires either (a) at least $q$ queries, or (b) a query of tolerance at most $\mu$, where $\min(q, 1/\mu^2) = d^{\Omega(\log^{0.99} d)}$. In particular, this holds even if $\mathrm{opt} \leq 1/k + \zeta + 1/k^3$.*

Given $k \in \mathbb{Z}_+$ and $k \geq 3$, we construct the corresponding noise matrix $H$ as

$$H = \begin{pmatrix} \frac{k-1}{k} - \zeta & 0 & \cdots & \frac{1}{k} + \zeta \\ 0 & \frac{k-1}{k} - \zeta & \cdots & \frac{1}{k} + \zeta \\ \cdots & \cdots & \cdots & \cdots \\ \frac{1}{k} - \frac{\zeta}{k-1} & \frac{1}{k} - \frac{\zeta}{k-1} & \cdots & \frac{1}{k} + \zeta \end{pmatrix}.$$

Recall that for a hidden direction distribution family $\mathcal{D}$, $\mathrm{err}_D(k) = 1 - 1/k - \zeta$, no matter if $D = D_0$ or $D \in \mathcal{D}$. Thus, if we can learn a hypothesis with error $1 - 1/k - \zeta - o(1)$, we are able to solve the correlation testing problem. To make this possible, we need to make $\mathrm{opt}$ as small as possible. By our construction, if an $x$ has ground truth label $f^*(x) \in [k-1]$, the probability that it is flipped is only $1/k + \zeta$. Thus, if we are able to choose $\sum_{j=1}^{k-1} \mathbf{Pr}(S_j) = \mathbf{Pr}(I_{in}) = 1 - 1/\mathrm{poly}(k)$, then $\mathrm{opt} = 1/k + \zeta + 1/\mathrm{poly}(k)$. By the tail bound of $N(0,1)$, to make this hold, we choose $m\delta = \Theta(\sqrt{\log k})$. Recall that we still need to make $m$, the degree of the polynomial we use, as small as possible. Here we choose $m = \Theta(\sqrt{N \log k})$, which suffices to give a super-polynomial SQ lower bound.

Given Theorem 6.2, we immediately obtain two corollaries for the hardness of approximate leaning and beating a random guess hypothesis respectively in the setting of multiclass linear classification. We defer the proofs of these two corollaries to Appendix D.3.

**Corollary 6.3** (SQ hardness of Approximate Learning). *For any $C > 1$, there exists a noise matrix $H \in [0,1]^{k \times k}$, with $k = O(C)$ and $\min_{i,j} H_{i,i} - H_{i,j} = \Omega(1/C)$ such that any SQ algorithm $\mathcal{A}$ that distribution-free learns multiclass linear classifiers on $\mathbb{R}^d$ with RCN parameterized by $H$ to error $C\mathrm{opt}$ given $\mathrm{opt} = \Omega(1/C)$ either*

*(a) requires at least $d^{\Omega(\log^{0.99} d)}$ queries, or*
*(b) requires a query of tolerance at most $1/d^{\Omega(\log^{0.99} d)}$.*

**Corollary 6.4** (SQ hardness of Beating Random Guess). *For any $k \in \mathbb{Z}_+$ and $k \geq 3$, there is a noise matrix $H \in [0,1]^{k \times k}$ with $\min_{i,j} H_{i,i} - H_{i,j} = 1/\mathrm{poly}(d)$ such that any SQ algorithm $\mathcal{A}$ that distribution-free learns multiclass linear classifiers on $\mathbb{R}^d$ with RCN parameterized by $H$ to error $1 - 1/k - 1/\mathrm{poly}(d)$ given $\mathrm{opt} = O(1/k)$ either*

*(a) requires at least $d^{\Omega(\log^{0.99} d)}$ queries, or*
*(b) requires a query of tolerance at most $1/d^{\Omega(\log^{0.99} d)}$,*

It is worth noting that if we predict the label of an example with $y \in [k]$ uniformly at random, then the error is always $1 - 1/k$. Therefore, Corollary 6.4 implies that it is not possible for an efficient SQ algorithm to output a hypothesis with error nontrivially better than a random guess hypothesis.

## 7. Conclusion

We conclude this paper with a conceptual implication of our results. Our SQ lower bounds exhibit the existence of a very simple multi-index model that is easy to learn with perfect labels, but is hard to learn even with a small level of random label noise. Finally, we remark that the results of our work motivate several interesting directions, including the algorithmic study of MLC with more structured noise or structured marginal distributions.

## Acknowledgement

Ilias Diakonikolas was supported by NSF Medium Award CCF-2107079 and an H.I. Romnes Faculty Fellowship. Mingchen Ma and Christos Tzamos were supported by NSF Award CCF-2144298 (CAREER). Lisheng Ren was supported in part by NSF Medium Award CCF-2107079.

## Impact Statement

This work is theoretical in nature and focuses on advancing fundamental knowledge. As such, it does not directly raise any societal or ethical concerns that warrant special consideration.

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

## Supplementary Material

Here we give an organization of the supplementary material. In Appendix A, we provide a complete list of preliminaries and notations and give more background about statistical query learning. In Appendix B, we present omitted proofs from Section 4. In Appendix C, we present omitted proofs from Section 5 and in Appendix D we give missing proofs in Section 6.

We want to remark that our SQ lower bound results have immediate implications to another well-studied restricted computational model—that of low-degree polynomial tests (see (Hopkins & Steurer, 2017; Hopkins et al., 2017; Hopkins, 2018)). (Brennan et al., 2020) established that (under certain assumptions) an SQ lower bound also implies a qualitatively similar lower bound in the low-degree model. This connection can be used as a black-box to deduce similar lower bounds for low-degree polynomials.

## A. Complete Preliminaries and Notations

Let $f^* : X \to Y$ be the ground truth hypothesis. For $j \in [k]$, denote by $S_j = \{x \mid f^*(x) = j\} \subseteq \mathbb{R}^d$ be the set of examples with $f^*(x) = j$. Let $h : X \to Y$ be an arbitrary hypothesis. For $i, j \in [k]$, we denote by $S_{ji} = \{x \mid f^*(x) = j, h(x) = i\} \subseteq \mathbb{R}^d$, the set of examples with ground truth label $j$, but on which $h$ predicts $i$. In this paper, we use $\mathbb{S}^{d-1}$ to denote the unit sphere in $\mathbb{R}^d$. Let $K \subseteq \mathbb{R}^d$ be any set, we denote by **conv**$(K)$, the convex hull of $K$. For a noise matrix $H \in [0, 1]^{k \times k}$, we denote by $h_i, i \in [k]$, the $i$th row vector of $H$.

For a distribution $D$, we use $\mathbf{E}_{\mathbf{x} \sim D}(x)$ to denote the expectation of $D$. Let $D$ be a distribution of $(x, y)$ over $\mathbb{R}^d \times [k]$. We use $D_X$ to denote the marginal distribution of $D$ over $\mathbb{R}^d$ and use $D_y$ to denote the marginal distribution of $D$ over $\pm 1$. In this paper, we will use $N(0, I)$ to denote the standard Gaussian distribution over $\mathbb{R}^d$ and use $N(0, 1)$ to denote the standard one-dimensional normal distribution. For $N(0, 1)$, we use $G(x)$ to denote its density function and use $\gamma_t, t \in \mathbb{N}$ to denote its standard $t$th moment $\mathbf{E}_{x \sim N(0,1)} x^t$.

**Definition A.1** (SQ Model). Let $D$ be a distribution over $X \times Y$. A *statistical query* is a bounded function $q : X \times Y \to [-1, 1]$. We define STAT$(\tau)$ to be the oracle that given any such query $q$, outputs a value $v$ such that $|v - \mathbf{E}_{(x,y) \sim D}[q(x, y)]| \leq \tau$, where $\tau > 0$ is the *tolerance* parameter of the query. A *statistical query (SQ) algorithm* is an algorithm whose objective is to learn some information about an unknown distribution $D$ by making adaptive calls to the corresponding STAT$(\tau)$ oracle.

**Definition A.2** (Pairwise Correlation). The pairwise correlation of two distributions with probability density function $D_1, D_2 : \mathbb{R}^d \mapsto \mathbb{R}_+$ with respect to a distribution with density $D : \mathbb{R}^d \mapsto \mathbb{R}_+$, where the support of $D$ contains the support of $D_1$ and $D_2$, is defined as $\chi_D(D_1, D_2) := \int_{\mathbb{R}^d} D_1(\mathbf{x})D_2(\mathbf{x})/D(\mathbf{x})d\mathbf{x} - 1$. Furthermore, the $\chi$-squared divergence of $D_1$ to $D$ is defined as $\chi^2(D_1, D) := \chi_D(D_1, D_1)$.

**Definition A.3** (Statistical Query Dimension). For $\beta, \gamma > 0$, a decision problem $\mathcal{B}(\mathcal{D}, D)$, where $D$ is a fixed distribution and $\mathcal{D}$ is a family of distribution, let $s$ be the maximum integer such that there exists a finite set of distributions $\mathcal{D}_D \subseteq \mathcal{D}$ such that $\mathcal{D}_D$ is $(\gamma, \beta)$-correlated relative to $D$ and $|\mathcal{D}_D| \geq s$. The Statistical Query dimension with pairwise correlations $(\gamma, \beta)$ of $\mathcal{B}$ is defined to be $s$, and denoted by $s = \text{SD}(\mathcal{B}, \gamma, \beta)$. We say that a set of $s$ distribution $\{D_1, \cdots, D_s\}$ over $\mathbb{R}^d$ is $(\gamma, \beta)$-correlated relative to a distribution $D$ if $\chi_D(D_i, D_j) \leq \gamma$ for all $i \neq j$, and $\chi_D(D_i, D_j) \leq \beta$ for $i = j$.

**Lemma A.4** ((Feldman et al., 2017a)). *Let $\mathcal{B}(\mathcal{D}, D)$ be a decision problem, where $D$ is the reference distribution and $\mathcal{D}$ is a class of distribution. For $\gamma, \beta > 0$, let $s = \text{SD}(\mathcal{B}, \gamma, \beta)$. For any $\gamma' > 0$, any SQ algorithm for $\mathcal{B}$ requires queries of tolerance at most $\sqrt{\gamma + \gamma'}$ or makes at least $s\gamma'/(\beta - \gamma)$ queries.*

**Fact A.5** (Fact 31 from (Diakonikolas & Kane, 2022)). *For any constant $0 < c < 1/2$, there exists a set $V \subseteq \mathbb{S}^{d-1}$ such that $|V| = 2^{\Omega_c(d)}$ and for any $u, v \in V$, $|u \cdot v| \leq c$.*

## B. Omitted Proofs from Section 4

In this section, we provide the omitted proofs in Section 4.

### B.1. Proof of Lemma 4.3

We provide the full proof of Lemma 4.3 here and restate Lemma 4.3 as Lemma B.1 for convenience.

**Lemma B.1.** *Let $(D, f^*, H)$ be any instance of multiclass classification with RCN. Let $h : X \to Y$ be an arbitrary*

*multiclass hypothesis over $X$. Then,*

$$\mathrm{err}(h) = \sum_{j=1}^{k} \mathbf{Pr}(S_j)(1 - H_{jj}) + \sum_{i \neq j} \mathbf{Pr}(S_{ji})(H_{jj} - H_{ji}).$$

*In particular, if $H_{jj} - H_{ji} \geq 0$ for every $j \in [k], i \neq j$, then*

$$\mathrm{opt} = \mathrm{err}(f^*) = \sum_{j=1}^{k} \mathbf{Pr}(S_j)(1 - H_{jj})$$

*Proof of Lemma 4.3.* For $j, i \in [k]$, let $x \in A_{ji}$ be any fixed example. Consider two cases, where $j = i$ and $j \neq i$. In the first case, we have

$$\mathbf{Pr}(h(x) \neq y(x)) = \mathbf{Pr}(y(x) \neq j) = 1 - H_{jj}.$$

In the latter case, we have

$$\mathbf{Pr}(h(x) \neq y(x)) = \mathbf{Pr}(y(x) \neq i) = 1 - H_{ji}.$$

Thus, we have

$$\mathrm{err}(h) = \sum_{j=1}^{k} \sum_{i=1}^{k} \mathbf{Pr}(S_{ji}) \mathbf{Pr}(h(x) \neq y(x) \mid x \in S_{ji}) = \sum_{j=1}^{k} \sum_{i=1}^{k} \mathbf{Pr}(S_{ji})(1 - H_{ji})$$

$$= \sum_{j=1}^{k} \sum_{i \neq j} \mathbf{Pr}(S_{ji})(1 - H_{ji}) + \left( \mathbf{Pr}(S_j) - \sum_{i \neq j} \mathbf{Pr}(S_{ji}) \right)(1 - H_{jj}) \tag{2}$$

$$= \sum_{j=1}^{k} \mathbf{Pr}(S_j)(1 - H_{jj}) + \sum_{j=1}^{k} \sum_{i \neq j} \mathbf{Pr}(S_{ji})(H_{jj} - H_{ji}).$$

Here, in the second equation, we use the fact that $\mathbf{Pr}(S_{jj}) = \mathbf{Pr}(S_j) - \sum_{i \neq j} \mathbf{Pr}(S_{ji}), \forall j \in [k]$. Since $\mathbf{Pr}(S_{ji}) \geq 0, \forall j, i \in [k]$, we know from (2) that $\mathrm{err}(h) \geq \sum_{j=1}^{k} \mathbf{Pr}(S_j)(1 - H_{jj})$, if $H_{ii} - H_{ji} \geq 0, \forall i, j \in [k]$. In particular, by (2), $\mathrm{err}(f^*) = \sum_{j=1}^{k} \mathbf{Pr}(S_j)(1 - H_{jj})$, which concludes

$$\mathrm{opt} = \mathrm{err}(f^*) = \sum_{j=1}^{k} \mathbf{Pr}(S_j)(1 - H_{jj}).$$

$\square$

## B.2. Proof of Lemma 4.4

We present the full proof of Lemma 4.4. We restate Lemma 4.4 as Lemma B.2

**Lemma B.2.** *Let $\mathcal{D}$ be a family of distribution over $X \times Y$ such that each distribution $D \in \mathcal{D}$ corresponds to a multiclass classification instance $(D, f^*, H)$ that satisfies Definition 4.1. If there is a statistical query learning algorithm $\mathcal{A}$ such that for every instance $(D, f^*, H), D \in \mathcal{D}$, $\mathcal{A}$ makes $q$ queries and each of them has tolerance $\tau$, and outputs a hypothesis $\hat{h}$ such that $\mathrm{err}(\hat{h}) \leq \mathrm{opt} + \alpha$, where*

$$2\alpha = \sum_{j=1}^{k-1} \mathbf{Pr}(S_j)(H_{jj} - H_{jk}),$$

*then there is a statistical learning algorithm $\mathcal{A}'$ that solves the correlation testing problem defined in Definition 4.2 by making $q + 1$ queries and each of them has error tolearance $\min(\tau, \alpha/2)$.*

*Proof of Lemma 4.4.* The algorithm $\mathcal{A}'$ works as follows. We run $\mathcal{A}$ over $D$ to get a hypothesis $\hat{h} : X \to Y$. Given $\hat{h}$, we make one more statistical query $q$ to estimate $\mathrm{err}(\hat{h})$ with tolerance $\alpha/2$. Denote by $\hat{\mathrm{err}}(\hat{h})$ the returned answer of $q$. We reject the null hypothesis if $\hat{\mathrm{err}}(\hat{h}) < 1 - H_{kk} - \alpha/2$ and accept the null hypothesis otherwise. The SQ complexity of the algorithm directly follows its definition. In the rest of the proof, we prove its correctness.

If $D = D_0$, since the label $y$ is drawn independently from $x$ and $\mathbf{Pr}(y = k) = H_{kk} \geq \mathbf{Pr}(y = j) = H_{kj}, \forall j \neq k$, any hypothesis $h : X \to Y$ has

$$\mathrm{err}(h) \geq \mathrm{err}(k) = \mathbf{Pr}(y \neq k) = 1 - H_{kk}.$$

This implies that, $\hat{\mathrm{err}}(\hat{h}) \geq 1 - H_{kk} - \alpha/2$ and $\mathcal{A}'$ will not reject the null hypothesis.

In the rest of the proof, we will show that if $D \in \mathcal{D}$, the algorithm $\mathcal{A}'$ will reject the null hypothesis. To start with, we will show that opt is $2\alpha$ far from $1 - H_{kk}$. On the one hand, by Lemma 4.3, we have

$$
\begin{aligned}
\mathrm{err}(k) &= \sum_{j=1}^{k} \mathbf{Pr}(S_j)(1 - H_{jj}) + \sum_{j=1}^{k-1} \mathbf{Pr}(S_{jk})(H_{jj} - H_{jk}) \\
&= \sum_{j=1}^{k-1} \mathbf{Pr}(S_j)(1 - H_{jk}) + \mathbf{Pr}(S_k)(1 - H_{kk}) \\
&= (1 - \mathbf{Pr}(S_k)) \sum_{j=1}^{k-1} \frac{\mathbf{Pr}(S_j)}{(1 - \mathbf{Pr}(S_k))}(1 - H_{jk}) + \mathbf{Pr}(S_k)(1 - H_{kk}) \\
&= (1 - \mathbf{Pr}(S_k)) \sum_{j=1}^{k-1} a_j(1 - H_{jk}) + \mathbf{Pr}(S_k)(1 - H_{kk}) = \sum_{j=1}^{k} \mathbf{Pr}(S_j)(1 - H_{kk}) = 1 - H_{kk}.
\end{aligned}
$$

Here, the second equation holds because by the definition of the constant hypothesis $S_j = S_{jk}, \forall j \in [k]$ and the fourth and the fifth equations are followed by Definition 4.1. On the other hand, by Lemma 4.3, $\mathrm{opt} = \sum_{j=1}^{k} \mathbf{Pr}(S_j)(1 - H_{jj})$. Thus,

$$1 - H_{kk} - \mathrm{opt} = \mathrm{err}(k) - \mathrm{opt} = \sum_{j=1}^{k} \mathbf{Pr}(S_j)(1 - H_{jj}) + \sum_{j=1}^{k-1} \mathbf{Pr}(A_{jk})(H_{jj} - H_{jk}) - \sum_{j=1}^{k} \mathbf{Pr}(S_j)(1 - H_{jj}) = 2\alpha,$$

which gives us that $\mathrm{opt} = 1 - H_{kk} - 2\alpha$. Given any hypothesis $\hat{h}$ output by $\mathcal{A}$ with $\mathrm{err}(\hat{h}) \leq \mathrm{opt} + \alpha$, we have $\mathrm{err}(\hat{h}) \leq 1 - H_{kk} - \alpha$. Thus, $\hat{\mathrm{err}}(\hat{h}) \geq 1 - H_{kk} - 2\alpha/2$ and $\mathcal{A}'$ will reject the null hypothesis. This concludes the proof of Lemma 4.4.

$\square$

# C. Omitted Proofs from Section 5

## C.1. Proof of Lemma 5.4

In this section, we present the proof of Lemma 5.4. For convenience, we restate Lemma 5.4 as Lemma C.1.

**Lemma C.1** (Distribution Projection). *Let $\mathcal{D}$ be a hidden direction distribution family over $\mathbb{R}^d$ and let $D_v^{A,a} \in \mathcal{D}$ be a distribution that is consistent with an instance of multiclass polynomial classification with RCN $(D_v^{A,a}, f^*, H)$. For every $i \in [k]$,*

$$D_v^{A,a}(x \mid y = i) = \sum_{j=1}^{k-1} \frac{a_j H_{ji}}{H_{ki}} P_v^{A_j}(x).$$

*Proof of Lemma 5.4.* We consider the density function of $D_v^{A,a}$ at a fixed point $(x, i)$,

$$D_v^{A,a}(x, i) = \sum_{j=1}^{k-1} a_j P_v^{A_j}(x) H_{f^*(x)i}$$

By Definition 5.2, we consider two cases for $x$. In the first case, $v \cdot x \in J_\ell$ for some $\ell \in [k-1]$. By construction of the distribution family $A = (A_1, \ldots, A_{k-1})$, $P_v^{A_j}(x) = 0, \forall j \neq \ell$. Thus,

$$D_v^{A,a}(x,i) = \sum_{j=1}^{k-1} a_j P_v^{A_j}(x) H_{\ell i} = a_\ell P_v^{A_\ell}(x) H_{\ell i} + \sum_{j \neq \ell} a_j P_v^{A_j}(x) H_{ji} = \sum_{j=1}^{k-1} a_j P_v^{A_j}(x) H_{ji}.$$

In the second case, $v \cdot x \notin I_{in}$ and thus $f^*(x) = k$. In this case,

$$D_v^{A,a}(x,i) = \sum_{j=1}^{k-1} a_j P_v^{A_j}(x) H_{ki} = P_v^{A_1}(x) H_{ki} = P_v^{A_1}(x) \sum_{j=1}^{k-1} a_j H_{ji} = \sum_{j=1}^{k-1} a_j P_v^{A_j}(x) H_{ji}.$$

Here, the second and the last equation holds because $P_v^{A_j}(x)$ is the same for every $j \in [k]$. The third equation holds because $h_k = \sum_{j=1}^{k-1} h_j$. Since $(D_v^{A,a}, f^*, H)$ satisfies Definition 4.1, we know that $\mathbf{Pr}(y = i) = H_{ki}$. Thus, $\forall x \in \mathbb{R}^d$,

$$D_v^{A,a}(x \mid y = i) = \sum_{j=1}^{k-1} \frac{a_j H_{ji}}{H_{ki}} P_v^{A_j}(x).$$

$\square$

## C.2. Proof of Theorem 5.3

In this section, we present the proof of Theorem 5.3. For convenience, we restate Theorem 5.3 as follows.

**Theorem C.2.** *Let $\mathcal{B}(D_0, \mathcal{D})$ be a correlation testing problem, where $(D_0)_X = N(0, I)$ and $\mathcal{D}$ is a hidden direction distribution family. Suppose there exists some $\nu > 0$ such that for $\ell \leq t \in \mathbb{Z}_+$, the family of one-dimensional distribution $A_1, \ldots, A_{k-1}$ satisfies $\left| \mathbf{E}_{x \sim A_i} x^\ell - \gamma_\ell \right| \leq \nu$. Then,*

1. *every distribution $D_v^{A,a} \in \mathcal{D}$ is consistent with an instance of multiclass polynomial classification with RCN $(D_v^{A,a}, f^*, H)$ with degree at most $2m$ that satisfies Definition 4.1.*

2. *for any small enough constant $c > 0$, let $\beta = \max_{i,j} \chi_{N(0,1)}(A_i, A_j)$ and let $\tau := \nu^2 + c^t \beta$. Any statistical query algorithm that solves $\mathcal{B}(D_0, \mathcal{D})$ must make a query with accuracy better than $2\sqrt{\tau}$ or make $2^{\Omega_c(d)} \tau / \beta$ queries.*

*Proof of Theorem 5.3.* We first prove Item 1 in Theorem 5.3.

Since for each $i \in [k-1]$, $J_i$ is a set of $m$ disjoint intervals, we know there is a degree-$2m$ polynomial $p_i(t) : \mathbb{R} \to \mathbb{R}$ such that $p_i(t) > 0$ if and only if $t \in J_i$. On the other hand, since $I_{in} = \mathbf{conv} \bigcup_{j \in [k-1]} J_j$ is a finite interval, there is a degree-2 polynomial $p_k(t) : \mathbb{R} \to \mathbb{R}$ such that $p_k(t) > 0$ if and only if $t \notin I_{in}$. Since $J_i \cap J_j = \emptyset, \forall i \neq j$. We know that for each $j \in [k-1]$, if $v \cdot x \in J_j$, then $j = \arg\max\{p_1(v \cdot x), \ldots, p_k(v \cdot x)\}$ and if $v \cdot x \notin I_{in}$, $k = \arg\max\{p_1(v \cdot x), \ldots, p_k(v \cdot x)\}$. In particular, $\mathbf{Pr}(v \cdot x \in I_{in} \setminus \bigcup_j J_j) = 0$ by the construction of the hidden direction distribution family. Thus, $D_v^{A,a}$ is consistent with an instance of multiclass polynomial classification with RCN $(D_v^{A,a}, f^*, H)$, where the marginal distribution is $\sum_{j=1}^{k-1} a_j P_v^{A_j}$ and the ground truth hypothesis $f^*(x) = \arg\max\{p_1(v \cdot x), \ldots, p_k(v \cdot x)\}$. In particular, by the definition of $(D_v^{A,a}, f^*, H)$, it satisfies Definition 4.1.

Next, we prove Item 2 in Theorem 5.3. Our proof strategy is to make use of Lemma A.4. To do this, we will bound $\chi_{D_0}(D_v^{A,a}, D_u^{A,a})$ for $v, u \in S$ for a pair of unit vectors $u, v$. For convenience, we mention the following lemma that will be used in the proof.

**Lemma C.3** (Lemma 13 in (Diakonikolas & Kane, 2022)). *Suppose there exists some $\nu > 0$ such that for $\ell \leq t \in \mathbb{Z}_+$, a univariate distribution $A$ satisfies $\left| \mathbf{E}_{x \sim A} x^\ell - \gamma_\ell \right| \leq \nu$, then for every $u, v \in \mathbb{R}^d$, with $|u \cdot v|$ less than a sufficiently small constant, we have*

$$\chi_{N(0,I)}(P_v^A, P_u^A) \leq |u \cdot v|^t \chi^2(A, N(0,1)) + \nu^2.$$

We start by upper-bounding the pairwise correlation $\chi_{D_0}(D_v^{A,a}, D_u^{A,a})$.

By Lemma 5.4, we know that for each $i \in [k]$,

$$D_v^i = \sum_{j=1}^{k-1} \frac{a_j H_{ji}}{H_{ki}} P_v^{A_j}(x) = P_v^{\sum_{j=1}^{k-1} \frac{a_j H_{ji}}{H_{ki}} A_j}(x)$$

. Since for each $j \in [k-1]$, $\left| \mathbf{E}_{x \sim A_j} x^\ell - \gamma_\ell \right| \leq \nu$ and $\sum_{j=1}^{k-1} \frac{a_j H_{ji}}{H_{ki}} = 1$, we know that,

$$\left| \underset{x \sim \sum_{j=1}^{k-1} \frac{a_j H_{ji}}{H_{ki}} A_j}{\mathbf{E}} x^\ell - \gamma_\ell \right| \leq \nu,$$

for $\ell \leq t$. Thus, we obtain

$$\chi_{D_0}(D_v^{A,a}, D_u^{A,a}) = \sum_{i=1}^{k} H_{ki} \chi_{D_0|y=i} \left( D_v^{A,a}(x \mid y = i), D_u^{A,a}(x \mid y = i) \right) = \sum_{i=1}^{k} H_{ki} \chi_{N(0,I)} \left( D_v^i, D_u^i \right)$$

$$\leq \sum_{i=1}^{k} H_{ki} \left( \nu^2 + |v \cdot u|^t \chi^2 \left( \sum_{j=1}^{k-1} \frac{a_j H_{ji}}{H_{ki}} A_j, N(0,1) \right) \right)$$

$$= \nu^2 + |v \cdot u|^t \sum_{i=1}^{k} H_{ki} \chi^2 \left( \sum_{j=1}^{k-1} \frac{a_j H_{ji}}{H_{ki}} A_j, N(0,1) \right)$$

$$\leq \nu^2 + |v \cdot u|^t \beta .$$

Here the first inequality holds because of Lemma C.3, and the last inequality holds as follows

$$\chi^2 \left( \sum_{j=1}^{k-1} \frac{a_j H_{ji}}{H_{ki}} A_j, N(0,1) \right) = \int_{-\infty}^{\infty} \frac{\sum_{j=1}^{k-1} \frac{a_j H_{ji}}{H_{ki}} A_j(x) \sum_{\ell=1}^{k-1} \frac{a_\ell H_{\ell i}}{H_{ki}} A_\ell(x)}{G(x)} dx - 1$$

$$= \sum_{j=1}^{k-1} \frac{a_j H_{ji}}{H_{ki}} \sum_{\ell=1}^{k-1} \frac{a_\ell H_{\ell i}}{H_{ki}} \left( \int_{-\infty}^{\infty} \frac{A_j(x) A_\ell(x)}{G(x)} dx - 1 \right)$$

$$= \sum_{j=1}^{k-1} \frac{a_j H_{ji}}{H_{ki}} \sum_{\ell=1}^{k-1} \frac{a_\ell H_{\ell i}}{H_{ki}} \chi_{N(0,1)}(A_i, A_j) \leq \beta.$$

Thus, for every $u, v \in \mathbb{S}^{d-1}$ such that $|u \cdot v| \leq c$, we have $\chi_{D_0}(D_v^{A,a}, D_u^{A,a}) \leq \nu^2 + c^{-k} \beta = \tau$.

Similarly, we upper bound $\chi^2 \left( D_u^{A,a}, N(0,I) \right)$ as follows.

$$\chi^2 \left( D_u^{A,a}, N(0,I) \right) = \sum_{i=1}^{k} H_{ki} \chi_{D_0|y=i} \left( D_v^{A,a}(x \mid y = i), D_v^{A,a}(x \mid y = i) \right)$$

$$= \sum_{i=1}^{k} H_{ki} \chi^2 \left( D_v^i, N(0,I) \right) = \sum_{i=1}^{k} H_{ki} \chi^2 \left( \sum_{j=1}^{k-1} \frac{a_j H_{ji}}{H_{ki}} A_j, N(0,1) \right) \leq \beta.$$

By Fact A.5, for any small constant $c > 0$, there exists a set $S$ of $2^{\Omega_c(d)}$ unit vectors such that for every $u, v \in S$, $|u \cdot v| \leq c$. Thus, $\mathrm{SD}(\mathcal{B}, \gamma, \beta) = 2^{\Omega_c(d)}$. By Lemma A.4, we conclude the proof.

$\square$

## C.3. Proof of Proposition 5.6

In this section, we present the full proof of Proposition 5.6. For convenience, we restate Proposition 5.6 as Proposition C.4.

**Proposition C.4.** *The univariant distributions $A_1, \ldots, A_{k-1}$ constructed in Definition 5.5 satisfies*

1. $\exists$ *a set of $m$ disjoint intervals $J_i, i \in [k-1]$ such that $A_i(x) > 0$, for $x \in J_i$ and $A_i(x) = 0$, for $x \in I_{in} \setminus J_i, I_{in} = \mathbf{conv} \bigcup_{j \in [k-1]} J_j$.*

2. $\forall x \in \mathbb{R} \setminus I_{in}, A_i(x) = A_j(x), \forall i, j \in [k-1]$.

3. *For $i, j \in [k-1], \chi_{N(0,1)}(A_i, A_j) \leq O(\delta/\xi)^2$.*

4. *For $t \in \mathbb{N}$ and for $i \in [k-1], |\mathbb{E}_{x \sim A_i} x^t - \gamma_t| \leq O(t!) \exp(-\Omega(1/\delta^2)) + 4(k-1)\xi(1 + 2m\delta)^t$.*

Before presenting the proof, it will be convenient to recall the following property proved by (Nasser & Tiegel, 2022).

**Fact C.5.** *For $\delta, \xi > 0$, $\left| \|G_{\delta,\xi}\|_1 - 1 \right| \leq \exp\left(-\Omega(1/\delta^2)\right), \left| 1/\|G_{\delta,\xi}\|_1 - 1 \right| \leq O(1)\exp\left(-\Omega(1/\delta^2)\right)$.*

*Proof of Proposition 5.6.* We first prove the first two properties. For $i \in [k-1]$, we define $J_i := \bigcup_{-m \leq n \leq m} [n\delta - (4i - 3)\xi, n\delta - (4i-5)\xi]$. Notice that $I_{in} = \mathbf{conv} \bigcup_{j \in [k-1]} J_j = [-m\delta - (4k-7)\xi, m\delta + \xi]$. By construction, $A_i(x) = A_j(x)$ if $x \notin I_{in}$. On the other hand, consider $x \in I_{in}$. For $i = 1$, and $x \in I_{in}, A_1(x) > 0$ if and only if $x \in J_1$. By construction for $i \in [k-1]$ and $x \in I_{in}, A_i(x) > 0$ if and only if $(x + 4(i-1))\xi \in J_1$, which is equivalent to $x \in J_i$. Since $\delta > 4(k-1)\xi$, we know that $J_i \cap J_j = \emptyset, \forall i \neq j$. This implies that for every $i \in [k-1]$, and $x \in I_{in} A_1(x) > 0$ if $x \in J_i$ and $A_i(x) = 0$ if $x \in J_j$.

We next prove the third property. It is convenient to mention the fact that $\chi^2(A_1(x), N(0,1)) \leq O(\delta/\xi)^2$, proved in (Nasser & Tiegel, 2022). For any pair of $i, j \in [k-1]$, we have

$$
\begin{aligned}
\chi_{N(0,1)}(A_i, A_j) &= \int_{-\infty}^{\infty} \frac{A_i(x)A_j(x)}{G(x)} dx - 1 = \int_{x \notin I_{in}} \frac{G^2(x)}{G(x)} dx + \int_{x \in I_{in}} \frac{A_i(x)A_j(x)}{G(x)} dx - 1 \\
&\leq \chi^2(A_1(x), N(0,1)) + \int_{x \in I_{in}} \frac{A_i(x)A_j(x)}{G(x)} dx \\
&= O(\frac{\delta}{\xi})^2 + \int_{x \in I_{in}} \frac{A_i(x)A_j(x)}{G(x)} dx.
\end{aligned}
$$

Notice that if $i \neq j$, then for each $x \in I_{in}, A_i(x)A_j(x) = 0$, which implies that

$$
\int_{x \in I_{in}} \frac{A_i(x)A_j(x)}{G(x)} dx = 0.
$$

It remains to consider the case where $i = j$. In this case, we have

$$
\begin{aligned}
\int_{x \in I_{in}} \frac{A_i^2(x)}{G(x)} dx &\leq 2 \sum_{0 \leq n \leq m} \int_{n\delta - (4i-3)\xi}^{n\delta - (4i-5)\xi} \frac{A_i^2(x)}{G(x)} dx = 2\left(\frac{\delta}{\xi}\right)^2 \frac{1}{\|G_{\delta,\xi}\|_1} \sum_{0 \leq n \leq m} \int_{n\delta - (4i-3)\xi}^{n\delta - (4i-5)\xi} \frac{G^2(x + 4(i-1)\xi)}{G(x)} dx \\
&= 2\left(\frac{\delta}{\xi}\right)^2 \frac{1}{\|G_{\delta,\xi}\|_1} \sum_{0 \leq n \leq m} \int_{n\delta - (4i-3)\xi}^{n\delta - (4i-5)\xi} \frac{1}{\sqrt{2\pi}} \exp\left(-\frac{(x + 4(i-1)\xi)^2}{2}\right) \exp((4(i-1)\xi)^2) \\
&\leq 2\left(\frac{\delta}{\xi}\right)^2 \frac{1}{\|G_{\delta,\xi}\|_1} \int_{x \in \mathbb{R}} \frac{1}{\sqrt{2\pi}} \exp\left(-\frac{(x + 4(i-1)\xi)^2}{2}\right) \exp((4(i-1)\xi)^2) \leq O(1)
\end{aligned}
$$

Here, the last inequality holds when $\xi \leq 1/k$. Thus, for $i, j \in [k-1], \chi_{N(0,1)}(A_i, A_j) \leq O(\delta/\xi)^2$.

Finally, we prove the last property. It is convenient to mention the fact that $|\mathbf{E}_{x \sim A_1} x^t - \gamma_t| \leq O(t!) \exp(-\Omega(1/\delta^2))$, proved in (Nasser & Tiegel, 2022), which implies that it suffices to upper bound $|\mathbf{E}_{x \sim A_1} x^t - \mathbf{E}_{x \sim A_i} x^t|$. We have

$$
\left| \mathbf{E}_{x \sim A_1} x^t - \mathbf{E}_{x \sim A_i} x^t \right| = \left| \int_{x \in I_{in}} x^t dA_1(x) - \int_{x \in I_{in}} x^t dA_i(x) \right| = \left| \int_{x \in I_{in}} x^t dA_1(x) - \int_{x \in I_{in}} (x - 4(i-1)\xi)^t dA_1(x) \right|
$$

$$
\leq \sup_{x \in I_{in}} \left( (x - 4(i-1)\xi)^t - x^t \right) \leq \sum_{\ell=1}^{t} \binom{t}{\ell} (4(k-1)\xi)^\ell x^{t-\ell} \leq 4(k-1)\xi \sum_{\ell=1}^{t} \binom{t}{\ell} x^{t-\ell}
$$

$$
\leq 4(k-1)\xi(1 + |x|)^t \leq 4(k-1)\xi(1 + 2m\delta)^t.
$$

This concludes the proof of Proposition 5.6. $\qquad\square$

## D. Omitted Proofs from Section 6

In this section, we provide the omitted proofs in Section 6.

### D.1. Proof of Theorem 6.1

We present the full proof of Theorem 6.1. For convenience, we restate Theorem 6.1 below.

**Theorem D.1.** *There is a matrix $H \in [0,1]^{3 \times 3}$ with $H_{ii} - H_{ij} \geq 0.1, \forall i \neq j \in [3]$ such that any algorithm $\mathcal{A}$ that distribution-free learns multiclass linear classifier with random classification noise specified by $H$ on $\mathbb{R}^d$ to error $\mathrm{opt} + \epsilon, \epsilon \in (0, 1)$ either*

*(a) requires at least $d^{\Omega(\log^{0.98}(d)/\epsilon^{1.98})}$ queries, or*

*(b) requires a query of accuracy at least $d^{-\Omega(\log^{0.98}(d)/\epsilon^{1.98})}$.*

*Proof.* Consider following noise matrix,

$$
H = \begin{pmatrix} 0.6 & 0 & 0.4 \\ 0 & 0.6 & 0.4 \\ 0.3 & 0.3 & 0.4 \end{pmatrix}.
$$

Notice that one can reduce learning polynomial classifiers with RCN to MLC with RCN using the Veronese mapping. Suppose we have an algorithm $\mathcal{A}$ for solving multiclass linear classification problems. Then given an input distribution $D$ of $(x, y)$ over $\mathbb{R}^N \times [k]$ consistent with an instance of multiclass degree-$m$ polynomial classification problem. We apply the Veronese mapping $V(x) := (x, 1)^{\otimes m}$ on $x$. The distribution of $(V(x), y)$ over $\mathbb{R}^{(N+1)^{O(m)}} \times [k]$ is consistent with an instance of MLC with RCN specified by $H$. Therefore, to get an SQ lower bound for MLC over $\mathbb{R}^d$, it suffices for us to give an SQ lower bound for learning degree-$m$ polynomial classifiers over $\mathbb{R}^N$, where $d = N^{O(m)}$.

Furthermore, by Lemma 4.4, to get the SQ lower bound for learning polynomial classifiers, it suffices for us to give an SQ lower bound on a corresponding testing problem. Therefore, we construct a distribution family $\mathcal{D}$ of joint distributions of $(x, y)$ over $\mathbb{R}^N \times [k]$ such that each distribution in $\mathcal{D}$ is consistent with a multiclass polynomial classification problem with RCN $(D, f^*, H)$ as required by Lemma 4.4.

The construct $\mathcal{D}$ as the hidden direction distribution family defined in Definition 5.2. We choose $A_1$ and $A_2$ as specified in Definition 5.5, $a = (1/2, 1/2)$. Since the noise matrix $H$ satisfies $h_3 = (h_1 + h_2)/2$, by Theorem 5.3, we know that each distribution $D$ is consistent with an instance of multiclass polynomial classification problem with degree-$O(m)$ with RCN specified by $H$. To make use of Theorem 5.3 to get an SQ lower bound, it remains to choose parameters for $A_1, A_2$ such that it is hard to solve $\mathcal{B}(D_0, \mathcal{D})$.

Fix any small enough constant $\epsilon > 0$. We choose the parameters $m, \delta$ such that $m\delta = \epsilon$. By Proposition 5.6, we know that for every $t \in \mathbb{N}$, we have

$$
\left| \mathbf{E}_{x \sim A_i} x^t - \gamma_t \right| \leq O(t!) \exp\left(-\Omega(1/\delta^2)\right) + 12\xi(1 + 2m\delta)^t
$$

$$
\leq O(1) \left( \exp\left(t \log(t) - \Omega(1/\delta^2)\right) + \xi \exp(2\epsilon t) \right).
$$

We choose $\delta = 1/\sqrt{N}, m = \epsilon\sqrt{N}$ and $\xi = \exp(-2N^{0.99})$. For any $t \leq N^{0.99}$, we have

$$\left| \mathop{\mathbf{E}}_{x \sim A_i} x^\ell - \gamma_\ell \right| \leq O\left(\exp\left(t\log(t) - \Omega(1/\delta^2)\right) + \xi\exp(t)\right)$$

$$\leq \exp(-\Omega(N)) + \exp(-2N^{0.99} + N^{0.99}) = \exp(-\Omega(N^{0.99})) =: \nu.$$

By Lemma 4.4, we know that any statistical query learning algorithm that learns $(D_v^{A,a}, f^*, H)$ up to error $\mathrm{opt} + \alpha$, where $\alpha = \frac{1}{2}\sum_{j=1}^{2}(H_{jj} - H_{j3})\mathbf{Pr}(S_j)$ can solve $\mathcal{B}(D_0, \mathcal{D})$. By the construction of $D_v^{A,a}$ and $H$,

$$\alpha = 0.1 \mathop{\mathbf{Pr}}_{x \sim A_1}(x \in I_{in}) = 0.1 \sum_{-m \leq n \leq m} \int_{n\delta-\xi}^{n\delta+\xi} \frac{\delta}{\xi} \frac{1}{\|G_{\delta,\xi}\|} G(x)dx$$

$$\geq \Omega(1) \sum_{-m \leq n \leq m} \int_{n\delta-\xi}^{n\delta+\xi} (\frac{\delta}{\xi})G(x)dx \geq \Omega(1) \sum_{-m \leq n \leq m} 2\xi(\frac{\delta}{\xi})G(2m\delta) \geq \Omega(1)(2m+1)\delta = \Omega(\epsilon).$$

Here, the first inequality holds because of Fact C.5, the second inequality holds because $G(x)$ is decreasing with respect to $|x|$ and the last inequality holds because $\epsilon = m\delta$. This implies that any statistical learning algorithm that learns the multiclass polynomial classification problem $(D_v^{A,a}, f^*, H), v \in \mathbb{S}^{N-1}$ up to error $\mathrm{opt} + O(\epsilon)$ must make at least $2^{\Omega(N)}$ statistical queries or a query with accuracy better than $\exp(-\Omega(N^{0.99}))$.

Finally, we conclude the proof of Theorem 6.1 by embedding the multiclass polynomial classification problem $(D_v^{A,a}, f^*, H), v \in \mathbb{S}^{N-1}$ into $\mathbb{R}^d, d = O(N^m)$ as a multiclass linear classification problem via choosing $m$ properly. By choosing $m = \epsilon\sqrt{N}$, we obtain that

$$\log(d) = \Theta(m\log(N)) = \Theta(\epsilon\sqrt{N}\log(N)).$$

any statistical learning algorithm that learns the multiclass linear classification problem over $\mathbb{R}^d$ up to error $\mathrm{opt} + O(\epsilon)$ must make at least

$$\exp(\Omega(N^{0.99})) = d^{\Omega(N^{0.99}/\log(d))} = d^{\tilde{\Omega}(\log^{0.98}(d)/\epsilon^{1.99})}$$

statistical queries or a query with accuracy better than $\exp(-\Omega(N^{0.99})) = d^{-\tilde{\Omega}(\log^{0.98}(d)/\epsilon^{1.99})}$.

$\square$

### D.2. Proof of Theorem 6.2

We give the proof of Theorem 6.2 below. For convenience, we state Theorem 6.2 as Theorem D.2.

**Theorem D.2.** *For any $k \in \mathbb{Z}_+$ and $k \geq 3$, there is a noise matrix $H \in [0,1]^{k \times k}$ such that $\max_{i,j} H_{i,i} - H_{i,j} = \zeta > 0$ and has the following property: For any sufficiently large $d \in \mathbb{Z}_+$, any algorithm $A$ that distribution-free learns multiclass linear classifier with random classification noise specified by $H$ on $\mathbb{R}^d$ to error $1 - 1/k - \zeta - 2\mu$ either*

*(a) requires at least $q$ queries, or*

*(b) requires a query of tolerance at most $\mu$,*

*where $\min(q, 1/\mu^2) = d^{\Omega(\log^{0.99} d)}$. In particular, this holds even if $\mathrm{opt} \leq 1/k + \zeta + 1/k^3$.*

*Proof of Theorem 6.2.* Given $k \in \mathbb{Z}_+$ and $k \geq 3$, we construct the corresponding noise matrix $H$ as

$$H = \begin{pmatrix} (k-1)/k - \zeta & 0 & \cdots & 1/k + \zeta \\ 0 & (k-1)/k - \zeta & \cdots & 1/k + \zeta \\ \cdots & \cdots & \cdots & \cdots \\ 1/k - \zeta/(k-1) & 1/k - \zeta/(k-1) & \cdots & 1/k + \zeta \end{pmatrix}.$$

Namely, for all $i \in [k-1]$, the $i$th row is defined as $H_{i,i} = (k-1)/k - \zeta$, $H_{i,k} = 1/k + \zeta$ and $H_{i,j} = 0$ for any $j$ such that $j \neq i$ and $j \neq k$. Then the $k$th row is defined as $H_{k,k} = 1/k + \zeta$ and $H_{k,j} = 1/k - \zeta/k$ for any $j \neq k$.

Given the noise matrix $H$, we will construct a hidden direction distribution family $\mathcal{D}$ over $\mathbb{R}^N$ that is consistent with a family of multiclass polynomial classification problems with RCN $(D^{A,a}, f^*, H)$ using polynomials of degree $O(m)$ for some $m$ to be determined later and prove the SQ hardness for these multiclass polynomial classification problems. Given the hardness, the hard instance of multiclass linear classification problems with RCN, would be an instance in $\mathbb{R}^d$, where $d = N^{O(m)}$ of the form $(M(x), y), (x,y) \sim D^{A,a}$, where $M(x) : \mathbb{R}^N \to \mathbb{R}^d$ defined as $M(x) = [x,1]^{\otimes m}$ is the degree-$m$ Veronese mapping that maps a vector $x \in \mathbb{R}^N$ to monomials of degree at most $m$.

To start with, we construct the hidden direction distribution family $\mathcal{D}$ using the distributions $A_1, \ldots, A_{k-1}$ constructed in Definition 5.5. Notice that $H$ satisfies Definition 4.1 because $h_k = \sum_{i \in [k-1]} a_i h_i$, where $a_i = 1/(k-1)$ for all $i$. We know from Theorem 5.3 that for every $A = (A_1, \ldots, A_{k-1})$, where $A_1, \ldots, A_{k-1}$ are one dimension distributions constructed in Definition 5.5, $\mathcal{D} = \{D_v^{A,a}\}_{v \in \mathbb{S}^{N-1}}$ is a hidden direction distribution family consistent with a family of multiclass polynomial classification problems with RCN $(D^{A,a}, f^*, H)$ using polynomials of degree $O(m)$. Now, we choose parameters for $A_1, \ldots, A_{k-1}$ such that it is hard to solve $\mathcal{B}(D_0, \mathcal{D})$.

We will choose $\delta = 1/\sqrt{N}, \xi = \exp(-N^{0.99} \log k)$ and $m = \lceil C\sqrt{\log k}/\delta \rceil$, where $C$ is a sufficiently large constant. By Proposition 5.6, we know that for every $t \in \mathbb{N}$, we have

$$\left| \mathop{\mathbf{E}}_{x \sim A_i} x^t - \gamma_t \right| \leq O(t!) \exp\left(-\Omega(1/\delta^2)\right) + 12\xi(1 + 2m\delta)^t$$
$$\leq O\left(\exp\left(t\log(t) - \Omega(1/\delta^2)\right) + \xi \exp(t\sqrt{\log k})\right).$$

Therefore, we get for any $t \leq N^{0.99}$,

$$\left| \mathop{\mathbf{E}}_{x \sim A_i} x^\ell - \gamma_\ell \right| \leq O\left(\exp\left(t\log(t) - \Omega(1/\delta^2)\right) + \xi \exp(t\sqrt{\log k})\right)$$
$$\leq \exp(-\Omega(N)) + \exp(-N^{0.99}\log k + N^{0.98}\sqrt{\log k}) = \exp(-\Omega(N^{0.99})) =: \nu.$$

By Proposition 5.6, we know that

$$\beta := \max_{i,j} \chi_{\mathcal{N}(0,1)}(A_i, A_j) = O(\delta/\xi)^2 = \exp(O(N^{0.99}\log k)).$$

This implies that (we take the constant $c$ in Lemma C.3 as $c^{-1} > 2^{10\log k}$)

$$\tau := \nu^2 + c^t\beta \leq \exp(-\Omega(N^{0.99})) + \exp(-\Omega(N^{0.99}\log k)) = \exp(-\Omega(N^{0.99})).$$

By Theorem 5.3, we know that to solve the correlation testing problem $\mathcal{B}(D_0, \mathcal{D})$, one need at least $2^{\Omega_k(N)}\tau/\beta = 2^{\Omega_k(N)}$ (given $N$ is at least a sufficiently large constant depending on $N$) statistical queries or a query with accuracy better than $2\sqrt{\tau} = \exp(-\Omega(N^{0.99}))$. Furthermore, since $N$ is at least a sufficiently large constant depending on $k$, the lower bound on the number of queries is $2^{\Omega_k(N)} \geq 2^{\Omega(N^{0.99})}$, where we simply take $N^{0.01} \geq c(k)$ and $c(k)$ is the constant factor in $2^{\Omega_k(N)}$ that depends on $k$.

Notice that by Lemma 4.3, for any $D \in \mathcal{D}$, we have

$$\text{opt} = (1/k + \zeta) \mathop{\mathbf{Pr}}_{t \sim G_{\delta,\xi}} [t \in [-m\delta, m\delta]] + ((k-1)/k - \zeta) \mathop{\mathbf{Pr}}_{t \sim G_{\delta,\xi}} [t \in (-\infty, -m\delta - \delta/2] \cup [m\delta + \delta/2, \infty)]$$
$$\leq (1/k + \zeta) + \mathop{\mathbf{Pr}}_{t \sim G_{\delta,\xi}} [t \in (-\infty, -m\delta - \delta/2] \cup [m\delta + \delta/2, \infty)]$$
$$\leq (1/k + \zeta) + 2 \sum_{i > m} \int_{i\delta - \xi}^{i\delta + \xi} G_{\delta,\xi}(t)dt$$
$$\leq (1/k + \zeta) + 2 \mathop{\mathbf{Pr}}_{t \sim \mathcal{N}(0,1)} [t \geq m\delta] = (1/k + \zeta) + 1/\text{poly}(k).$$

On the other hand, if the input distribution is $D_0$, then every hypothesis $h$ has an error

$$\text{err}_{D_0}(h) \geq 1 - (1/k + \zeta) = 1 - 1/k - \zeta.$$

Therefore, any algorithm for learning multiclass polynomial classification with RCN matrix $H$ and achieving an error better than $1 - 1/k - \zeta - 2\mu$ can be used to solve $\mathcal{B}(D_0, \mathcal{D})$ with one more query with accuracy $2\mu$. Thus, any such algorithm must either uses $q$ queries or a query of tolerance at most $\mu$, where $q = 1/\mu = 2^{\Omega(N^{0.99})}$.

Finally, we conclude the proof of Theorem 6.1 by embedding the multiclass polynomial classification problem $(D_v^{A,a}, f^*, H), v \in \mathbb{S}^{N-1}$ into $\mathbb{R}^d, d = O(N^m)$ as a multiclass linear classification problem and rewrite. Given $d = O(N^m)$, we get that $2^{\Omega(N^{0.99})} = d^{\Omega(N^{0.99}/m)} = d^{\Omega(N^{0.49}/(c\sqrt{\log k}))} \geq d^{\Omega(N^{0.45})} \geq d^{\Omega(m^{0.9})} \geq d^{\Omega((\log d)^{0.9})}$. Therefore, any statistical learning algorithm that learns the multiclass linear classification problem over $\mathbb{R}^d$ to error $1 - (1/k + \zeta) = 1 - 1/k - \zeta$ (even given $\mathrm{opt} \leq 1/k + \zeta + 1/\mathrm{poly}(k)$) must make at least $d^{\Omega((\log d)^{0.9})}$ statistical queries or a query with accuracy better than $d^{-\Omega((\log d)^{0.9})}$. $\square$

## D.3. Proof of Corollary 6.3 and Corollary 6.4

We present the proof of Corollary 6.3 and Corollary 6.4. For convenience, we restate Corollary 6.3 and Corollary 6.4 as Corollary D.3 and Corollary D.4 respectively.

**Corollary D.3** (SQ hardness of approximate learning). *For any $C > 1$, there exists a noise matrix $H \in [0, 1]^{k \times k}$, where $k = O(C)$ and $\min_{i,j} H_{i,i} - H_{i,j} = \Omega(1/C)$ that has the following property: For any $d \in \mathbb{Z}_+$ that is at least a sufficiently large constant depending on $\alpha$, any algorithm $A$ that distribution-free learns multiclass linear classifier on $\mathbb{R}^d$ with RCN parameterized by $H$ to error $C\mathrm{opt}$ given $\mathrm{opt} = \Omega(1/C)$ either*

(a) *requires at least $d^{\Omega(\log^{0.99} d)}$ queries, or*

(b) *requires a query of tolerance at most $1/d^{\Omega(\log^{0.99} d)}$.*

*Proof.* This directly follows from Theorem 6.2, where we take $k = \lceil 3C \rceil$ and $\zeta = 1/(100k)$. Then we have $\mathrm{opt} = 1/k + \zeta + 1/k^3 = 1.01/k + 1/k^3$. An algorithm that achieves error $\alpha\mathrm{opt}$ given $\mathrm{opt} = O(1/C)$ will in this case, output a hypothesis with error $\alpha\mathrm{opt} \leq (k/3)\mathrm{opt} \leq 2/3$. Notice that the SQ lower bound in Theorem 6.2 holds against any algorithm that outputs a hypothesis with error at most $1 - 1/k - \zeta - 1/\mathrm{poly}(d) = 1 - 1/k - 0.01/k - 1/\mathrm{poly}(d) \geq 2/3$. This completes the proof. $\square$

**Corollary D.4** (SQ hardness of beating random guess). *For any $k \in N$ and $k \geq 3$, there is a noise matrix $H \in [0, 1]^{k \times k}$ that $\min_{i,j} H_{i,i} - H_{i,j} = O(1/d)$ and has the following property: For any $d \in \mathbb{Z}_+$ that is at least a sufficiently large constant depending on $k$, any algorithm $A$ that distribution-free learns multiclass linear classifier on $\mathbb{R}^d$ with RCN parameterized by $H$ to error $1 - 1/k - 1/\mathrm{poly}(d)$ given $\mathrm{opt} = O(1/k)$ either*

(a) *requires at least $d^{\Omega(\log^{0.99} d)}$ queries, or*

(b) *requires a query of tolerance at most $1/d^{\Omega(\log^{0.99} d)}$,*

*Proof.* This directly follows from Theorem 6.2. Suppose that there is an algorithm achieving error $1 - 1/k - 1/d^c$ for any $c > 0$. Then we take $\zeta = 1/(2d^c)$. Given $d$ is a sufficiently large constant depending on $k$, it is easy to check that $\mathrm{opt} = 1/k + \zeta + 1/k^3 = 1/k + 1/(2d^c) + 1/k^3 = O(1/k)$. Furthermore, the SQ lower bound holds against any algorithm that outputs a hypothesis with error at most $1 - 1/k - \zeta - 1/\mathrm{poly}(d) = 1 - 1/k - \zeta - 2\tau \geq 1 - 1/k - 1/d^c$, where the last inequality follows from $\tau = d^{\Omega(\log d)^{0.99}}$. This completes the proof. $\square$

*Remark* D.5. We want to remark that in Corollary 6.4, to rule out an efficient learning algorithm that has a better error guarantee than $1 - 1/k$, any choice of $\zeta = o_k(1)$ is sufficient, and the separation $\min_{i \neq j} H_{ii} - H_{ij}$ of $H$ is in fact $O(\zeta)$.

