# OpenReview forum: "Statistical Query Hardness of Multiclass Linear Classification with Random Classification Noise"
_ICML.cc/2025/Conference — ICML 2025 oral_

### Official Review · Reviewer_4NDD · 2025-02-25

**Overall Recommendation:** 4

**Summary:**

This paper studies the problem of learning multiclass linear classifiers of form $
f_w(x) := \arg\max_{i} \\{\langle w_i, x \rangle\\},$
under random classification noises with known noise channel. The paper shows that, unlike the binary classification case where efficient SQ algorithm is possible,  the case with even 3 classes is SQ-hard with query complexity superpolynomial w.r.t. the input dimension.

The main proof technique is based on reducing the hardness to a composite hypothesis testing problem. The difficulty lies in constructing the hypothesis testing distributions consistenting with certain multiclass polynomial classifiers (which future reduces to linear classifiers with a larger input dimention). This is achieved by leveraging the hidden direction distributions from Diakonikolas & Kane (2022) and Nasser & Tiegel (2022), with a more careful design for the distributions on the "hidden direction". The final SQ lower bound then follows by tuning the degree of the polynomials.

**Claims And Evidence:**

The claims are clear, and the proofs are convincing.

**Essential References Not Discussed:**

N/A

**Experimental Designs Or Analyses:**

N/A

**Methods And Evaluation Criteria:**

This is a purely theoretical paper. The proof technique is sound and based on prior literature.

**Other Comments Or Suggestions:**

**Writing Suggestions:**

1. The notation $C\text{opt}$ is confusing, consider using $C \cdot \text{opt}$.
2. In Definition 4.1, item 2 should be $x \not\in S_k$.
3. Definition 5.2 is quite confusing:
   - it would be better to explicitly say that "$J_i$ is a union of $m$ disjoint intervals." I only understood this after reading your construction in Definition 5.5.
   - did you miss $\textbf{conv}$ in the definition of $I_{\text{in}}$? Moreover, it is better to write $I_{\text{in}}$ instead of $I_{in}$, as one might confuse "in" with $i \cdot n$.
4. The distributions $h_i$s are confusing with the hypothesis, consider using $H_j$s.

**Other Strengths And Weaknesses:**

**Other Weaknesses:**

1. Although this is clearly a solid theoretical paper with novel ideas, I'm skeptical about its broad impact on the community. It seems to me that the difficulty mainly arises from the use of 0-1 loss, which is known to be hard in many other settings.

2. A more practically relevant setting (as also pointed out by the authors) is to consider softmax regression,
   $
   p_w(i \mid x) = \frac{e^{\langle w_i, x\rangle}}{\sum_{j=1}^{k} e^{\langle w_j, x\rangle}},
   $
   and optimize under the log-loss. This is known to admit computationally efficient algorithms even in the adversarial online setting (which reduces to the batch setting through online-to-batch conversion); see:
   [https://arxiv.org/pdf/2110.03960](https://arxiv.org/pdf/2110.03960).

3. The proof techniques largely follow from Diakonikolas & Kane (2022) and Nasser & Tiegel  (2022), which is not particularly surprising.

**Questions For Authors:**

I list some comments in the "Other Strengths and Weaknesses" section, which will not impact my recommendation for acceptance.

**Relation To Broader Scientific Literature:**

This paper advances the understanding of the computational complexity of linear classification under RCN beyond binary labels.

**Theoretical Claims:**

I checked most of the proofs necessary for the main claims, though I didn't verify every single detail. As far as I can see, the overall proof idea is clean and correct.

---

> ### Author Rebuttal · Authors · 2025-04-01
>
> We thank the reviewer for appreciating our theoretical results and for providing useful feedback. We next respond to the comments from the reviewer as follows.
>
> >Classification v.s. Regression:
>
> We thank the reviewer for drawing our attention to the multi-class regression setting. In that respect, we want to point out that 0-1 loss is a standard benchmark for studying classification problems and for linear classification in particular. Specifically, the computational difficulty of the problem does not arise from using the 0-1 loss per se, but from the added noise. As we mentioned in the introduction (and also see our response to reviewer Y6KY), in the realizable setting the MLC problem can be written as a linear program and can therefore be solved efficiently (and with an SQ algorithm). Furthermore, if $H$ is well-conditioned, the problem is also efficiently solvable with respect to the 0-1 loss. In particular, the scale of the ground truth W has no effect on the 0-1 loss. That is, the use of the 0-1 loss is not the reason that makes the problem hard.
>
> On the other hand, the logistic regression problem studied in https://arxiv.org/pdf/2110.03960 is not comparable to the classification setting studied in our paper. In fact, under the logistic loss, even the ground truth hypothesis W* could have a very large loss in the realizable setting—while under the 0-1 loss in the realizable setting W* has 0-1 loss equal to 0. This means that, without reasonable distributional assumptions, we cannot hope to use logistic regression to learn a hypothesis with a good 0-1 loss, which is what we believe is most desirable in practice. Furthermore, the use of logistic regression depends on the magnitude of W, while in 0-1 loss the magnitude of W does not affect the learnability of the problem. For two W’s with the same prediction under 0-1 loss, their logistic losses can be very different. In fact, the regret bound obtained in https://arxiv.org/pdf/2110.03960 actually depends on this magnitude (which can blow up when W is large). Based on these observations, we believe that the results for logistic regression are somewhat orthogonal to our results and their future potential impact. In summary, we believe that the hardness results obtained in our paper may motivate subsequent algorithmic work under additional structural assumptions (please also see the conclusions of our paper).
>
> >Technical Novelty:
>
> Given the efficient learnability of halfspaces with RCN and MLC in the realizable setting, prior to our work, it was considered plausible that MLC with RCN is also efficiently learnable. As mentioned in the introduction, MLC with RCN has been widely studied in prior work both empirically and theoretically (by different communities), and efficient learning algorithms have been developed when $H$ is well-conditioned.
> Rather surprisingly, we show a strong computational separation between the binary and the ternary cases.
> Furthermore, we want to emphasize that although parts of our proof are built over the standard NGCA framework developed by [DKS17, DK22] and leverage modifications of the discrete Gaussian inspired by [NT22], formulating MLC to fit in these frameworks requires novel ideas, such as: identifying the correct condition, under which the hardness holds; and mapping MLC to a polynomial classification problem without using polynomials with very high degree. We refer the reviewer to our response to Y6KY’s similar question for more details.
>
>
> >Reference:
>
> [BPSTWZ19]Beygelzimer, A., Pal, D., Szorenyi, B., Thiruvenkatachari, D., Wei, C.-Y., and Zhang, C. Bandit multiclass linear classification: Efficient algorithms for the separable case. In International Conference on Machine Learning, pp. 624–633. PMLR, 2019.
>
> [DV04]Dunagan J, Vempala S. A simple polynomial-time rescaling algorithm for solving linear programs. InProceedings of the thirty-sixth annual ACM symposium on Theory of computing 2004 Jun 13 (pp. 315-320)
>
> [DKS17]Diakonikolas, I., Kane, D. M., and Stewart, A. Statistical query lower bounds for robust estimation of highdimensional gaussians and gaussian mixtures. In 2017 IEEE 58th Annual Symposium on Foundations of Computer Science (FOCS), pp. 73–84, 2017. doi: 10.1109/ FOCS.2017.16.
>
> [DK22]Diakonikolas, I. and Kane, D. Near-optimal statistical query hardness of learning halfspaces with massart noise. In Conference on Learning Theory, pp. 4258–4282. PMLR, 2022.
>
> [NT22]Nasser, R. and Tiegel, S. Optimal sq lower bounds for learning halfspaces with massart noise. In Conference on Learning Theory, pp. 1047–1074. PMLR, 2022.

---

### Official Review · Reviewer_Y6KY · 2025-03-08

**Overall Recommendation:** 4

**Summary:**

This paper is concerned with the task of multiclass linear classification (MLC) with $k$ labels under random classification noise (RCN). The problem parameters are a $k \times k$ row-stochastic noise matrix $H$, and a target linear classifier $f^\star$ that maps $\mathbb{R}^d$ to $[k]$ as $f^\star(x)=argmax_{i \in [k]}(w \cdot x)$ for some ground truth vector $w \in \mathbb{R}^d$. There is a joint distribution $D$ over $\mathbb{R}^d \times [k]$, such that a sample is drawn as follows: first, we draw $x \sim D_x$ where $D_x$ is the marginal of $D$ on $\mathbb{R}^d$, and then, we draw $y$ from the conditional distribution $\Pr[y=j|x]=H_{f^\star(x), j}$. Namely, the label $y$ should ideally have been $f^\star(x)$ with no noise; but now, we perturb the label according to $f^\star(x)^{\text{th}}$ row of $H$. Given iid draws from this distribution, and $\epsilon \in (0,1)$, the task in MLC with RCN is to output a classifier that has error at most $\epsilon$ larger than the error of the optimal classifier in some class.

The sample complexity of this task is known. This paper is concerned with the computational complexity of the task. In particular, for $k=2$, computationally efficient algorithms (that furthermore fit into the SQ model of algorithms) are known. One of the main results in the paper (Theorem 1.2) is an SQ lower bound for the case when $k=3$. This is surprising, and establishes a separation between what can be achieved computationally efficiently for $k=2$ versus $k=3$. The authors also establish certain other results, which rule out SQ-based algorithms to even approximate the optimal error upto a multiplicative constant factor (Theorem 1.3), and also an instance that is hard to output a hypothesis better than random guessing in an efficient manner (Theorem 1.4).

The results are accomplished by combining prior SQ-lower bound constructions in the literature due to Diakonikolas and others. In particular, the authors first show how a certain hypothesis testing problem that distinguishes cases where the labels are independent of $x$, versus the case where the labels are drawn according to a specialized MLC with RCN instance (see Definition 4.1), reduces to the learning problem of MLC with RCN (Lemma 4.4). So, the main task then becomes showing that this testing problem is hard. For this, the authors use a construction based on so-called "hidden direction distributions" (Definition 5.1) inspired from prior work. This construction appears to be standard in the SQ lower bound literature, but massaging it to the present context of MLC with RCN appears to require novel conceptual bridges. Finally, all the aforemenetioned hardness results for MLC under RCN follow (Section 6) from the hardness of distinguinging in the above testing problem.

## update after rebuttal
I thank the authors for the clarifications. It would be definitely be useful to include this discussion (as the authors deem appropriate) in the revised version of the paper. I maintain my evaluation of the paper, and my score.

**Claims And Evidence:**

The claims and evidence appear convincing to me.

**Essential References Not Discussed:**

NA

**Experimental Designs Or Analyses:**

NA

**Methods And Evaluation Criteria:**

NA

**Other Comments Or Suggestions:**

I would really encourage the authors to be more specific with particular lemmas/theorems/chapters when you refer to textbooks (e.g. the SSS-SBD textbook). Simply citing a textbook in my opinion is quite lazy and entirely useless.

**Other Strengths And Weaknesses:**

To me, the primary strength of the paper is the conceptual conclusion---within the class of computationally efficient algorithms, SQ algorithms suffice for binary classification with RCN, but provably do not for multiclass problems even with 3 classes. This conclusion is indeed intriguing to me. I believe this the conclusion would inspire future inquiry into the specifics of what can be achieved efficiently for MLC under RCN

To establish the result, the authors do rely on heavy-weight machinery on SQ lower bounds from prior literature. I do not necessarily view this as a weakness, since as i mentioned, massaging prior literatrure into the present context does appear to require technical work, and the final conclusion is satisfying and important.

**Questions For Authors:**

1) Could you, in a brief paragraph, summarize the techincal conceptual novelty that is required in massaging the standard hidden distribution SQ lower bounds to the present context? While the overview in section 2 is helpful, I was not able to fully appreciate what parts in the overview were specifically novel and challenging for the present result.

2) What is the reference to the LP result for the realizable settingin lines 35-37?

3) I am a little confused about what the difference even is in non-realizable vs realizable learning under RCN. Since the labels are always perturbed by noise, and the error is measured with respect to the best classifier in a class, what does it mean to be realizable? Does the non-realizable case simply refer to the case where $f^\star$ is not in the class, so opt is no longer $f^\star$? If so, for the binary case, what is known about the difference between these two settings?

**Relation To Broader Scientific Literature:**

A lot of statistical learning theory is concerned with analyzing the number of samples that are information-theoretically sufficient and necessary to output an accurate classifier. The computational complexity of generating these accurate classifiers however also demands study. The present result establishes a separation in the computational complexity of linear classification under noise when the number of labels increases from 2 to 3. This is conceptually a surprising result, and suggests that standard algorithmic paradigms (namely, SQ algorithms) do not suffice for the seemingly natural extension for learning tasks from binary to multiclass, and more powerful algorithmic primitives provably become necessary.

**Theoretical Claims:**

I only glanced over the proofs, and did not verify calculations line-by-line. They appear correct to me, and the progression in the overall analysis checks out to me.

---

> ### Author Rebuttal · Authors · 2025-04-01
>
> We thank the reviewer for the appreciation and useful feedback.
>
> >Technical Contributions:
>
> We point out that our proof cannot be viewed as a simple modification of a previously developed SQ lower bound. As explained in the submission, the generic framework follows the moment-matching approach for Non-Gaussian Component analysis of [DKS17] and its generalization [DK22]. More precisely, we require a “relativized” version of that framework that is appropriate for supervised learning. Importantly, in order to be able to use this framework for our learning task, we need to construct novel low-dimensional moment-matching distributions that correspond to instances of MLC with RCN. This is our main contribution that requires significant conceptual and technical work.
>
> Specifically, to achieve this, we carefully map an instance of MLC to a polynomial classification problem and derive a new “hardness” condition (Definition 4.1), under which (after the mapping) the conditional distribution of the polynomial classification problem coincides with a modified discrete Gaussian. Prior to our work, it was unclear how a hard instance of MLC with RCN would look like (or whether such a hard instance exists). While the use of the modified version of a discrete Gaussian is inspired by prior works (such as [NT22]), reaching this step requires novel technical insights for MLC with RCN.
> In particular:
>
> 1. A key novelty of our paper is to identify the “correct” condition (Definition 4.1) under which we are able to relate our problem with the existing framework for proving SQ lower bounds. Identifying such a condition is highly non-trivial for the following reasons. Conceptually speaking, algorithms developed in prior works fail when the noise matrix $H$ is non-invertible. However, not all non-invertible matrices $H$ can be used to obtain hard instances. Specifically, for a generic non-invertible $H$, we can only guarantee that one row of $H$ can be written as a linear combination of the other rows of $H$; and the coefficients of the linear combination could be negative. Natural attempts to construct hard instances for a generic $H$ fail. Specifically, they may require mapping MLC to a polynomial classification problem with very high degree, which will not lead to a super-polynomial lower bound for the original problem (as explained in Section 6). On the other hand, using our SQ hard-to-distinguish condition (Definition 4.1), we are able to map MLC with RCN to a polynomial classification problem in a clean and novel way (Theorem 5.3); together with the modification of a discrete Gaussian (Definition 5.5) these rule out any efficient SQ learning algorithm for this problem.
>
> 2.
> Additionally, recognizing the hardness of MLC with RCN itself is already rather surprising, given the well-known efficient learners for halfspaces with RCN; and learning MLC in the realizable setting. Before our work, it seemed plausible that a similar result could hold for MLC with RCN, at least when $f^*$ is Bayes optimal. In fact, prior works have shown that when $H$ is well-conditioned, this is indeed possible. Our results settle the complexity of the problem by showing a surprising computational separation between MLC with RCN and binary classification with RCN.
>
> >LP formulation for MLC in the realizable setting:
>
> This reduction follows the definition of MLC. Instead of considering $k$ vectors in $R^d$, we can consider one vector $w$ in $R^{kd}$ (the concatenation of $w_1,..,w_k$) such that for every example $x$ with label $i$, we have $w_i.x-w_j.x \ge 0$ for every $j \neq i$ (see Definition 1 of [BPSTWZ19] for example).
>
> >Realizable v.s. Non-realizable:
>
> From the phrasing in the review, we understood that the question is the following: what is the difference between the case where $f^*$ is the Bayes optimal classifier (realizable) and the case where $f^*$ is not the Bayes optimal classifier (non-realizable). In case we misunderstood your question, please let us know and we will respond to the clarification.
>
> If the probability that the label of an example is unchanged is strictly larger than the probability that it is flipped to another label, then $f^*$ always achieves 0-1 error opt and is the Bayes optimal classifier. In the binary classification setting, this condition on the error probabilities is equivalent to saying that the probability the label of an example is flipped is less than $1/2$. On the other hand, if this condition does not hold, then $f^*$ may not be the hypothesis in the class that achieves error opt. However, in both cases, we always want to compete with the hypothesis in the class that achieves the optimal error (we do not want to compete with hypotheses not in the class). In binary classification, no matter whether $f^*$ achieves error opt or not, we are always able to learn a hypothesis with an error arbitrarily close to opt in polynomial time.
>
> The references are included in the response to Reviewer 4NDD.

---

### Official Review · Reviewer_8DF6 · 2025-03-12

**Overall Recommendation:** 4

**Summary:**

The paper studies linear multiclass classification under random classification noise within the statistical query (SQ) model. It considers a setting with positive noise separation, meaning that for a given labeled pair $(x, f^{\star}(x))$ under the ground truth labeling hypothesis, the probability of correctly observing the label $y = f^{\star}(x)$ is strictly greater than the probability of observing any incorrect label $y \neq f^{\star}(x)$. The paper establishes a superpolynomial lower bound on the number of queries required for any SQ learner. Additionally, when the number of labels is large, it provides a superpolynomial lower bound for approximate SQ learners or SQ learners that outperform random guessing.

**Claims And Evidence:**

Possible issue with Theorem 6.2. See theoretical claims below.

**Essential References Not Discussed:**

N/A

**Experimental Designs Or Analyses:**

N/A

**Methods And Evaluation Criteria:**

N/A

**Other Comments Or Suggestions:**

N/A

**Other Strengths And Weaknesses:**

I think this is a strong paper that establishes several interesting results. And the paper does a good job of conveying the intuition behind some of the constructions used in their lower bound proofs.

As for potential weaknesses, given that this appears to be the first work studying linear multiclass classification with random noise from an efficiency perspective, it would have been valuable to include some positive results as well. See the questions below for further discussion on this point. But this is just a personal opinion, so I won't take this into account in my judgement of the paper.

**Questions For Authors:**

*(This question is only relevant if the concern I raised earlier about the proof has a minor fix.)*

In the introduction, the authors mention prior work proposing methods whose time complexity scales inverse polynomially with the minimum singular value of $H$. In their intuitive discussion of the proof, they describe a matrix $H$ in Equation (1) with linearly dependent columns and a separation of $\sigma = 0.1$. Based on this example, it seems that such a large separation is feasible only for $k = 3$, and as $k$ increases the separation has to grow progressively smaller for the lower bound to hold.


This suggests a possibility of a phase transition. Specifically, for each $k$, there might exist a threshold $\sigma$—a function of $k$—such that if the separation is at least $\sigma$, a polynomial-time bound is achievable. For sufficiently large $\sigma$, it might be possible to establish a fixed lower bound on the smallest singular value in terms of $\sigma$, which, combined with the existing results cited in the introduction, would imply the existence of an efficient algorithm.

**Relation To Broader Scientific Literature:**

There is a growing body of research showing that the statistical landscape of multiclass learning can differ significantly from binary classification, especially as the number of labels increases. This paper contributes to this literature by highlighting a key difference that also considers the computational landscape.

**Theoretical Claims:**

Yes, I went through Theorem 6.2, Corollary 6.3, and Corollary 6.4 at a high level.

I may be overlooking something, but I think there might be an issue in Theorem 6.2. Specifically, the sum of the entries in the $k$-th row of the noise matrix $H$ does not seem to add up to $1$. Instead, it sums to

$$ 1+ \zeta - \frac{(k-1)}{k} \zeta = 1+\frac{\zeta}{k}. $$

This defines a valid matrix only if $\zeta = 0$. I believe the intended definition of $H$ was

$$ H_{kj} = \frac{1}{k} -\frac{\zeta}{k-1} \quad \forall j < k, \quad \text{ and } \quad H_{kk} = \frac{1}{k} + \zeta. $$

I briefly reviewed the proof of Theorem 6.2 and did not see any immediate step where division by $\zeta$ would cause the proof to break. However, given that the proof involves asymptotic bounds with $O(\cdot)$ and $\Omega(\cdot)$ notation, it is unclear whether some of the hidden constants depend on $1/\zeta$. I would appreciate it if the authors could verify whether this issue has a straightforward fix or if it presents a more fundamental problem.

At the very least, it is evident that Corollaries 6.3 and 6.4 do not hold for $\zeta = 0$, as those proofs explicitly assume $\zeta$ is a nonzero constant.





$\textbf{Given this issue, I am recommending a rejection. However, I am happy to engage with the authors during the rebuttal and change my recommendation if a minor fix is provided. }$

---

> ### Author Rebuttal · Authors · 2025-04-01
>
> We would like to thank the reviewer for appreciating our theoretical results and for pointing out typos in the manuscript. We next respond to the comments from the reviewer as follows.
>
> >Definition of $H$ in the proof of Theorem 6.2:
>
> We want to thank the reviewer for pointing out this typo in the submission. The correct definition is as follows: the last row of H is defined as $H_{ki} = 1/k - \zeta/(k-1)$ for $i<k$ and $H_{kk} = 1/k+\zeta$.
> We will fix this minor issue in the revised version of the manuscript.
>
> Importantly, we want to emphasize that this typo does not affect the correctness of Theorem 6.2 and its corollaries for the following reasons.
>
> First, for any noise matrix $H$ that satisfies the SQ-hard to distinguish condition (Definition 4.1), we construct a family of hypothesis testing problems in a black-box way. The lower bound on the running time (number of SQ queries) for such a hypothesis-testing problem only depends on the parameters of the hard distributions defined for this hypothesis testing problem (Theorem 5.3). The choice of the parameter $\zeta$ only affects the final error guarantee that can be ruled out by our hardness result—but does not affect the lower bound on running time for the corresponding hypothesis testing problem. So, even setting $\zeta=0$ does not affect any proof; moreover, the big-O notation does not have any hidden dependence on $1/\zeta$. Specifically, in Theorem 5.3, the lower bound is $\Omega_c(N) = C(c) N$, where $C(c)$ is a number that only depends on the parameter $c$ used to calculate $\tau$ in the statement of Theorem 5.3. Using Theorem 5.3 we prove Theorem 6.2. Importantly, in doing this, no matter which $\zeta$ we choose in the definition of $H$, we always choose the constant $c=1/poly(k)$. This gives us an SQ lower bound of $2^{\Omega_k(N)}>2^{\Omega(N^{0.99})}$ when $N^{0.01}$ is larger than $C(k)$ in the hidden constant. In summary, the proof of the lower bound is correct as is. Second, as we mentioned earlier, the choice of $\zeta$ only affects the final error guarantee that we can rule out. In particular, either opt or the error guarantee is calculated with respect to $1-H_{ii}$, as we did in line 1031 and line 1043. Luckily, despite the typos for $H_{ki}$ for i<k in the definition of H, the values of the diagonal elements $H_{ii}$ are correct. In summary, the final error guarantee we rule out remains correct as is.
>
> >Possible positive results on MLC with RCN:
>
> We first want to point out that although our paper is the first paper that establishes a computational hardness result for MLC with RCN, as mentioned in the introduction of our paper, there exist many prior works that study this problem both empirically and theoretically from an algorithmic perspective. Specifically, when the noise matrix H is invertible, one can invert the noise matrix H and reduce the problem back to MLC in the realizable setting. These methods work well if H is well-conditioned; for example, when $H_{ii}>2/3$, inverting $H$ is easy and will not affect the learning task by much. However, these methods fail if H is ill-conditioned; in fact, we show that in the distribution-free setting the problem is provably computationally hard. As we point out in the conclusion section, there are several interesting directions that can be explored with respect to positive results. These include MLC with RCN under well-behaved marginal distributions, and MLC with RCN for more structured noise matrices. Positive results in these directions are beyond the scope of this paper.
>
> >Phase transition for $\sigma$:
>
> We start by pointing out the following: since the choice of $\sigma=0.1$ gives a hard noise matrix for $k=3$, one can easily obtain a hard noise matrix with $\sigma=0.1$ for any $k>3$. Specifically, this can be done by concatenating this $H$ (for $k=3$) with an identity matrix with size $k-3$. That is, it is not the case that the separation needs to be small in order to make the hardness result work for large $k$. On the other hand, we believe that there is a threshold $\sigma$ such that this phase transition happens. For example, if $\sigma=2/3$, then $H$ is well-conditioned (i.e., has no small eigenvalues), and (as was pointed out in prior work and mentioned above)
> we can invert $H$ to get a computationally efficient algorithm. However, the actual running time of such algorithms scales polynomially with the inverse of the smallest singular value of $H$ instead of the separation parameter. It would be interesting to understand this phase transition phenomenon more deeply
> and precisely characterize this threshold–namely, whether $H$ is not invertible for some $\sigma$, but we are still able to solve the problem efficiently.

---

> > ### Comment · Reviewer_8DF6 · 2025-04-02
> >
> > I thank the authors for the clarification regarding the typo. While I did not verify every detail line by line, the explanation provided seems reasonable. I also appreciate the clarification on the phase transition.
> >
> > Since my concerns have been addressed, I have raised my score from 1 to 4.

---

### Decision · Program_Chairs · 2025-05-01

**Decision:**

Accept (oral)

**Comment:**

This paper studies the fundamental problem of robustly learning multiclass linear models in the presence of random classification noise. It is shown that even for a problem with three classes, PAC learning is computationally hard by any SQ algorithm. In addition, such SQ-hardness remains even when we relax the PAC guarantee to constant approximation. All together, these results are significant for understanding the multiclass PAC learning problem. The paper is clearly written.